# Amide-to-ester substitution as a stable alternative to *N*-methylation for increasing membrane permeability in cyclic peptides

Yuki Hosono[1], Satoshi Uchida [1], Moe Shinkai[1], Chad E. Townsend[2], Colin N. Kelly[2], Matthew R. Naylor [2], Hsiau-Wei Lee[2], Kayoko Kanamitsu[3], Mayumi Ishii[3], Ryosuke Ueki [1], Takumi Ueda [3], Koh Takeuchi [3], Masatake Sugita[4,5], Yutaka Akiyama [4,5] ✉, Scott R. Lokey[2] ✉, Jumpei Morimoto [1] ✉ & Shinsuke Sando [1,6] ✉

Naturally occurring peptides with high membrane permeability often have ester bonds on their backbones. However, the impact of amide-to-ester substitutions on the membrane permeability of peptides has not been directly evaluated. Here we report the effect of amide-to-ester substitutions on the membrane permeability and conformational ensemble of cyclic peptides related to membrane permeation. Amide-to-ester substitutions are shown to improve the membrane permeability of dipeptides and a model cyclic hexapeptide. NMR-based conformational analysis and enhanced sampling molecular dynamics simulations suggest that the conformational transition of the cyclic hexapeptide upon membrane permeation is differently influenced by an amide-to-ester substitution and an amide *N*-methylation. The effect of amide-to-ester substitution on membrane permeability of other cyclic hexapeptides, cyclic octapeptides, and a cyclic nonapeptide is also investigated to examine the scope of the substitution. Appropriate utilization of amide-to-ester substitution based on our results will facilitate the development of membrane-permeable peptides.

Cyclic peptides are emerging as an attractive class of molecules for clinical applications. Recent developments in screening technology have led to the discovery of many cyclic peptide inhibitors targeting challenging proteins[1–3]. In addition, macrocyclization endows peptides with proteolytic resistance, improving their stability in the bloodstream. Despite these beneficial aspects, cyclic peptides generally show low membrane permeability, making it challenging to develop orally bioavailable peptides and peptides targeting intracellular

protein–protein interactions[4,5]. The development of strategies to improve membrane permeability is important for expanding the utility of cyclic peptides.

The membrane permeability of peptides is governed by several physicochemical properties, such as molecular weight, the number of hydrogen bond donors and acceptors, and the polar surface area[6,7]. Previous research has shown that the number of hydrogen bond donors exposed to solvent in a lipophilic environment is one of the

[1]Department of Chemistry and Biotechnology, Graduate School of Engineering, The University of Tokyo, 7-3-1 Hongo, Bunkyo-ku, Tokyo 113-8656, Japan. [2]Department of Chemistry and Biochemistry, University of California, Santa Cruz, CA 95064, USA. [3]Graduate School of Pharmaceutical Sciences, The University of Tokyo, 7-3-1 Hongo, Bunkyo-ku, Tokyo 113-0033, Japan. [4]Department of Computer Science, School of Computing, Tokyo Institute of Technology, 2-12-1 Ookayama, Meguro-ku, Tokyo 152-8550, Japan. [5]Middle-Molecule IT-based Drug Discovery Laboratory (MIDL), Tokyo Institute of Technology, 2-12-1 Ookayama, Meguro-ku, Tokyo 152-8550, Japan. [6]Department of Bioengineering, Graduate School of Engineering, The University of Tokyo, 7-3-1 Hongo, Bunkyo-ku, Tokyo 113-8656, Japan. ✉e-mail: akiyama@c.titech.ac.jp; slokey@ucsc.edu; jmorimoto@chembio.t.u-tokyo.ac.jp; ssando@chembio.t.u-tokyo.ac.jp

most important factors governing the membrane permeability of peptides[7]. Peptides with large numbers of exposed amide NH groups usually show lower lipophilicity and passive membrane permeability because they have high desolvation penalties.

Some natural cyclic peptides exhibit high membrane permeability and oral bioavailability. These naturally occurring cyclic peptides often bear *N*-methylamide bonds, ester bonds, or both on their backbones. Cyclosporin A (CSA)[8], hirsutellide A[9,10], and guangomide[11] are representative examples of such natural peptides. Both *N*-methylamides and esters are planar and have similar bond lengths and angles to amides, but have no hydrogen bond donors. The absence of hydrogen bond donors is expected to reduce the desolvation energy. Inspired by these natural products, researchers have shown that *N*-methylation of backbone amide NH groups exposed in a lipophilic environment is a useful strategy to improve the membrane permeability of cyclic peptides by reducing the desolvation penalty for membrane permeation[4,12–16].

In contrast to backbone *N*-methylation, the effect of amide-to-ester substitutions on the membrane permeability of peptides has never been directly evaluated. Peptides with ester bonds on their backbones are often found in natural products, as are peptides with *N*-methylamides. The prevalence of these so-called depsipeptides with high membrane permeability suggests that the ester bond is favorable for achieving efficient membrane permeation[7]. There are also depsipeptides that can be retained in a membrane[17], which also suggests the high membrane-associating capability of depsipeptides. Moreover, the ester bond is an amide isostere with a similar bond length, cis–trans

propensity, and energy landscape on the Ramachandran diagram[18–21]. The effect of amide-to-ester substitutions on protein folding and peptide conformations has been investigated previously[22–26]. However, the effect of the substitution on peptides' permeability has not been directly evaluated. A previous study showed that a reverse ester-to-amide substitution on the backbone of a bioactive peptide did not affect its inhibitory activity in vitro but significantly reduced its inhibitory activity in cells[27], indirectly suggesting that the ester-to-amide substitution reduces the membrane permeability of this peptide. Another study showed that the octanol–water distribution coefficient, log $D_{o/w}$, of a peptide is increased by amide-to-ester substitution, indicating that the substitution increases the lipophilicity of the peptide. However, the permeability of the peptides was not evaluated in the study[28].

In this study, we directly compare the membrane permeabilities of peptides and their corresponding depsipeptides. We show the utility of the amide-to-ester substitution of peptides for improving peptide membrane permeability. In addition, the conformational analysis is conducted on a cyclic peptide with an amide-to-ester substitution on an exposed amide to evaluate the effect of the substitution on peptide conformations. Moreover, enhanced sampling molecular dynamics (MD) simulations are conducted to obtain insights into a plausible permeation mechanism of the cyclic depsipeptide. The scope of amide-to-ester substitution on membrane permeability is also examined using other cyclic hexapeptides and cyclic peptides with larger ring sizes.

## Results

### The effect of amide-to-ester substitution on permeability of model dipeptides

We first investigated how amide-to-ester substitutions affect the membrane permeability of peptides using dipeptides as model compounds. We synthesized a series of dipeptides (**P1–3**) and their derivatives containing an amide-to-ester substitution (**D1–3**) and an amide *N*-methylation (**M1–3**) (Fig. 1a, b). The sequences of the peptides were Ac-Xaa$_1$-Xaa$_2$-NH$_2$, where Ac denotes an acetylated N-terminus. The residues were linked by an amide in **P1–3**, by an ester in **D1–3**, and by an *N*-methylamide in **M1–3**. The two residues were Phe or Leu, making the lipophilicity reasonably high, thus allowing for facile detection of the peptides on permeability assays[29]. Passive permeability of individual peptides was evaluated using parallel artificial membrane permeability assay (PAMPA)[30,31]. For all the sequences, the depsipeptides exhibited the highest membrane permeability, which is at least a 10-fold enhancement compared with those of the corresponding peptides (Fig. 1c). Notably, the depsipeptides showed higher membrane permeability than the corresponding *N*-methylated peptides.

The higher membrane permeability of **D1–3** compared with **P1–3** can be attributed to the higher lipophilicity of ester compared with amide. Calculated distribution coefficients (CLogP and ALogP), retention time on octadecyl column during liquid chromatography, and experimental distribution coefficients between decadiene and aqueous buffer (log $D_{dec/w}$) were determined to assess the lipophilicity of the compounds (Supplementary Table 1). All the values of **D1–3** were higher than those of **P1–3**. The calculated LogP values (ALogP) and retention time on the octadecyl column of **D1–3** were also higher than those of **M1–3** although the log $D_{dec/w}$ values of **D1–3** were not very different from those of **M1–3**. These results indicate that ester is more lipophilic than amide and can be also more lipophilic than *N*-methylamide.

### The effect of amide-to-ester substitution on permeability of a cyclic peptide

To investigate whether amide-to-ester substitutions also enhance the membrane permeability of cyclic peptides, we next measured the membrane permeability of a cyclic hexapeptide (**CP1**) (Fig. 2a) and its

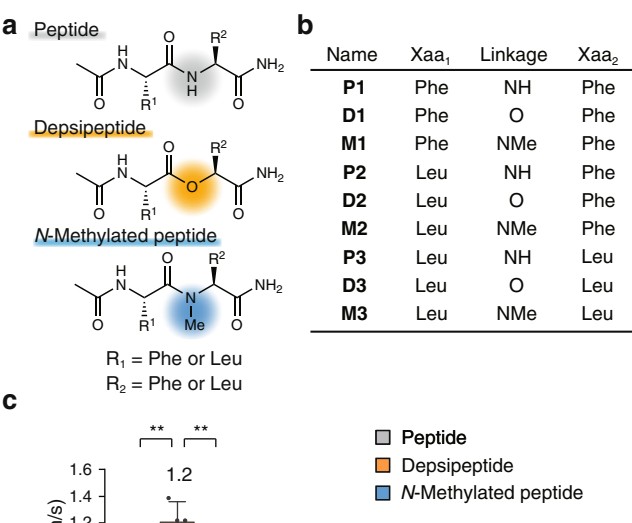

**b**

| Name | Xaa$_1$ | Linkage | Xaa$_2$ |
|------|---------|---------|---------|
| **P1** | Phe | NH | Phe |
| **D1** | Phe | O | Phe |
| **M1** | Phe | NMe | Phe |
| **P2** | Leu | NH | Phe |
| **D2** | Leu | O | Phe |
| **M2** | Leu | NMe | Phe |
| **P3** | Leu | NH | Leu |
| **D3** | Leu | O | Leu |
| **M3** | Leu | NMe | Leu |

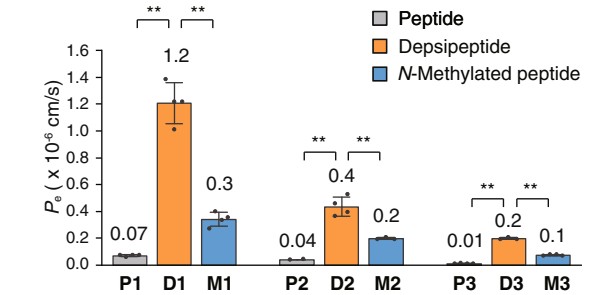

**Fig. 1 | The effect of amide-to-ester substitution on the permeability of dipeptides. a** General structures of model dipeptides. **b** Sequences of synthesized dipeptides. **c** Permeability values of the synthesized dipeptides measured by PAMPA. PAMPA was conducted with 10 µM compounds in 5% DMSO/PBS (pH 7.4) and 18 h incubation at 25 °C. Each bar represents the mean value, and the error bars the standard deviation from experiments carried out in quadruplicate. For **P2**, the peptides from the acceptor wells in two out of four trials were under the quantification limit, therefore the bar represents the mean value, and the error bars the standard deviation from experiments carried out in duplicate. *P* values were determined by a two-sided Welch's *t*-test. **$p < 0.01$. $p$ (**P1** vs. **D1**) = 0.0007, $p$ (**D1** vs. **M1**) = 0.0007, $p$ (**P2** vs. **D2**) = 0.0017, $p$ (**D2** vs. **M2**) = 0.0072, $p$ (**P3** vs. **D3**) < 0.0001 and $p$ (**D3** vs. **M3**) < 0.0001.

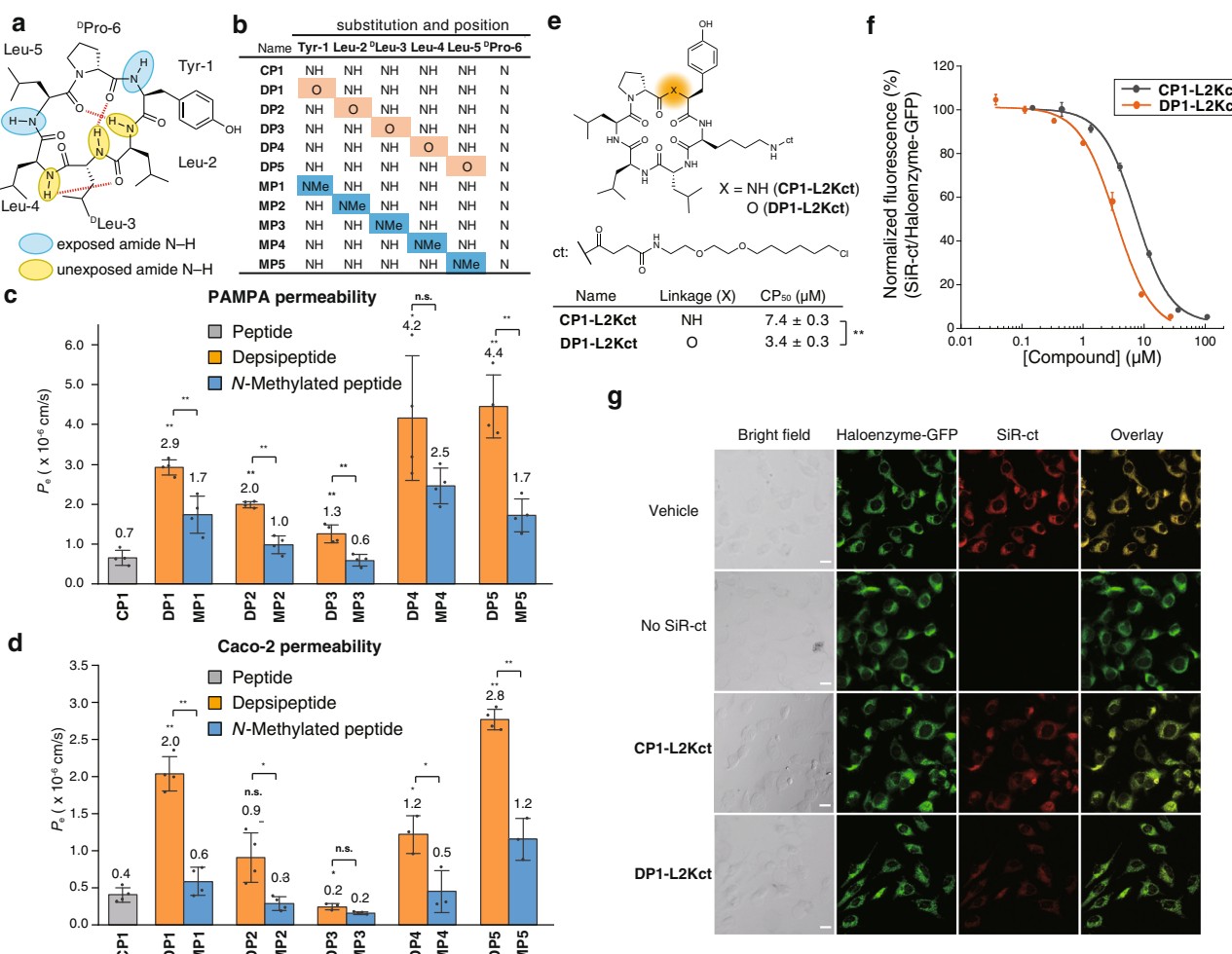

**Fig. 2 | The effect of amide-to-ester substitution on the permeability of a cyclic hexapeptide. a** Chemical structure of **CP1**. **b** A table of synthesized compounds. The position of an amide-to-ester substitution and amide *N*-methylation is shown by O highlighted in orange and NMe highlighted in blue, respectively. **c** PAMPA and **d** Caco-2 assay of synthesized cyclic peptides. PAMPA was conducted with 2 μM compounds in PBS containing 5% DMSO and 16 h incubation at 25 °C. Cyclosporin A (CSA) was included as a control for PAMPA ($0.4 \times 10^{-6}$ cm/s). Each bar represents the mean value, and the error bars the standard deviation from experiments carried out in quadruplicate. Caco-2 assay was conducted with 1 μM compounds in HBSS (pH 7.4) containing 10 mM HEPES and 1% DMSO and 3 h incubation at 37 °C. Each bar represents the mean value, and the error bars the standard deviation from experiments carried out in triplicate (**DP3**, **DP4**, **MP4**, and **MP5**) or quadruplicate (other than **DP3**, **DP4**, **MP4**, and **MP5**). The statistical significance of **DP1–5** against **CP1** is shown above the bar of **DP1–5** and the statistical significance of **DP1–5** against **MP1–5** is shown above the bars of **DP1–5** and **MP1–5**. *P* values were determined by a two-sided Welch's *t*-test. \*\**p* < 0.01, \**p* < 0.05. n.s. denotes not significant. *p* (**CP1** vs. **DP1**) < 0.0001, *p* (**DP1** vs. **MP1**) = 0.0092, *p* (**CP1** vs. **DP2**) =

0.0002, *p* (**DP2** vs. **MP2**) = 0.0014, *p* (**CP1** vs. **DP3**) = 0.0070, *p* (**DP3** vs. **MP3**) = 0.0037, *p* (**CP1** vs. **DP4**) = 0.0197, *p* (**DP4** vs. **MP4**) = 0.1148, *p* (**CP1** vs. **DP5**) = 0.0016, and *p* (**DP5** vs. **MP5**) = 0.0024 for PAMPA. *p* (**CP1** vs. **DP1**) = 0.0002, *p* (**DP1** vs. **MP1**) < 0.0001, *p* (**CP1** vs. **DP2**) = 0.0511, *p* (**DP2** vs. **MP2**) = 0.0293, *p* (**CP1** vs. **DP3**) = 0.0406, *p* (**DP3** vs. **MP3**) = 0.0689, *p* (**CP1** vs. **DP4**) = 0.0214, *p* (**DP4** vs. **MP4**) = 0.0253, *p* (**CP1** vs. **DP5**) < 0.0001, and *p* (**DP5** vs. **MP5**) = 0.0039 for Caco-2 assay. **e** Chemical structure, linkages, and $CP_{50}$ values of chloroalkane-tagged cyclic peptides. $CP_{50}$ values, the concentrations at which 50% cell penetration was observed, are shown at the bottom. **f** The results of CAPA for **CP1-L2Kct** (gray) and **DP1-L2Kct** (orange) analyzed by flow cytometry. Each data point represents the mean value of experiments carried out in triplicate and the error bars represent standard deviations of the triplicate. **g** Confocal microscope images of cells in CAPA. The cells were treated with 5 μM peptide solution. Green fluorescence represents a fusion protein of GFP and HaloTag, and red fluorescence represents SiR-ct dye. A scale bar (20 μm) is included in the bright field image of each dataset. The experiment was repeated with minor modifications and a similar result was obtained (Supplementary Fig. 9).

derivatives with amide-to-ester substitutions (**DP1–5**) and backbone *N*-methylations (**MP1–5**) (Fig. 2b). We adopted **CP1** as a model because previous studies showed that **CP1** has a low but detectable membrane permeability, and its stable conformations have been well studied[13,32,33]. The stable conformations of **CP1** were determined in a previous conformational study using NMR spectroscopic analysis in CDCl₃, which has a similar dielectric constant to that in the center of the membrane. According to this conformation, **CP1** has two exposed amide NHs at Tyr-1 and Leu-5 in a lipophilic environment (Fig. 2a). **DP1**, **MP1**, **DP5**, and **MP5** are peptides with an amide-to-ester substitution or an amide *N*-methylation at one of these potentially exposed amide NHs in the membrane.

The membrane permeability of the cyclic peptides was initially evaluated using PAMPA (Fig. 2c). All the five peptides with amide-to-ester substitutions exhibited significantly higher membrane permeabilities ($P_e = 2.9 \times 10^{-6}$, $2.0 \times 10^{-6}$, $1.3 \times 10^{-6}$, $4.2 \times 10^{-6}$, and $4.4 \times 10^{-6}$ cm/s for **DP1–5**, respectively) than **CP1** ($P_e = 0.7 \times 10^{-6}$ cm/s) (Fig. 2c). Notably, four of the five depsipeptides (**DP1**, **DP2**, **DP3**, and **DP5**) have higher permeabilities than their corresponding *N*-methylated analogs. The permeability enhancement of **DP1** and **DP5** indicates that the membrane permeability of a peptide can be improved by introducing an amide-to-ester substitution on an exposed amide bond. Unexpectedly, amide-to-ester substitutions at unexposed amides (Leu-2, D-Leu-3, and Leu-4) also improved permeability, probably because the substitutions led to the

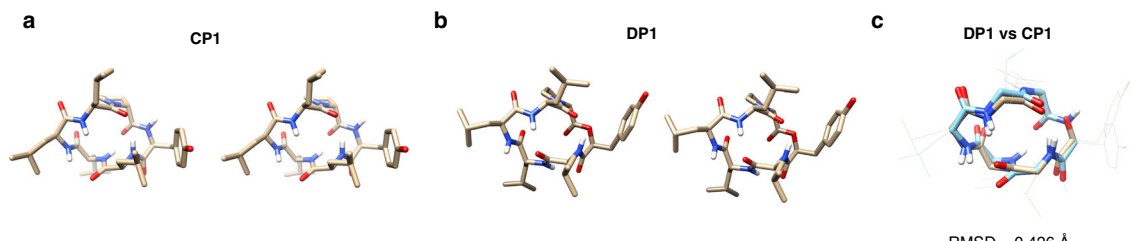

**Fig. 3 | NMR solution structures of CP1 and DP1.** Stereoviews of NMR solution structures of **a** CP1 and **b** DP1 in CDCl₃. **c** The superposition of DP1 with CP1. DP1 is shown in brown and **CP1** is shown in blue. The root mean square deviation (RMSD) value is shown under the structures.

loss of hydrogen bonding networks, which in turn changed the conformational preferences of the cyclic peptides in lipophilic media. To verify the conformational changes, we studied solution conformations of one of the three peptides, **DP2**, using NMR spectroscopy (Supplementary Figs. 1–6). From the NMR spectra in CDCl₃, stable conformations of **DP2** in the lipophilic environment were determined. Interestingly, the amide NH of Tyr-1 residue in the most stable conformations of **DP2** forms an intramolecular hydrogen bond with the carbonyl oxygen of the Leu-5 residue (Supplementary Fig. 7a) while the same amide NH was reported to be solvent-exposed in **CP1**[32] (Supplementary Fig. 7b). The amide-to-ester substitution on the amide that forms an intramolecular hydrogen bond in **CP1** caused the rearrangement of the intramolecular hydrogen bonding network, which is assumed to be the reason for the unexpectedly improved membrane permeability of **DP2**. The unexpected improvement of membrane permeability of **DP3** and **DP4** is also probably due to the conformational changes upon the amide-to-ester substitution as seen in **DP2**.

To examine whether the amide-to-ester substitution strategy also works on living cells, the cell-membrane permeability of the cyclic peptides was measured using a Caco-2 assay (Fig. 2d). A trend similar to that of PAMPA permeability was observed for the permeability on the Caco-2 assay. **DP1** and **DP5** exhibited significantly higher permeability than **CP1**. Unlike the observation in PAMPA, **DP2** did not show a significant difference in permeability compared to **CP1** (Fig. 2d). However, when the same assay was conducted at a higher concentration (55 μM), **DP2** showed significantly higher permeability than **CP1**, which indicated the involvement of efflux transporters (Supplementary Fig. 8)[34]. The efflux ratio, $P_e$ (basolateral-to-apical)/$P_e$ (apical-to-basolateral), of **DP2** at 1 μM concentration was determined to be 27.2, confirming the involvement of efflux transporters (Supplementary Table 2). The permeability value of **DP2** was also measured in the presence of efflux transporter inhibitors (quinidine, sulfasalazine, and benzbromarone)[35]. The efflux ratio was reduced to 1.05 in the presence of the inhibitors (Supplementary Table 2), further confirming the involvement of efflux transporters and suggesting the involvement of one or more of the three major efflux transporters in the intestinal epithelium; P-glycoprotein (MDR1/P-gp; ABCB1), breast cancer resistance protein (BCRP; ABCG2) and multidrug-resistance-associated protein 2 (MRP2; ABCC2). **DP3** showed similar permeability to **CP1**, and **DP4** showed moderately higher permeability than **CP1** (Fig. 2d).

We also conducted a cell-membrane penetration assay called ChloroAlkane Penetration Assay (CAPA)[36,37] to confirm that the depsipeptides penetrate the cell membrane and translocate into the cytosol. We synthesized a **CP1** derivative and a **DP1** derivative with a chloroalkane tag in place of the side chain of Leu-2 (**CP1-L2Kct** and **DP1-L2Kct**, respectively) (Fig. 2e). The cells used in this assay are HeLa cells that stably express a fusion protein of HaloTag, GFP, and a mitochondrial targeting peptide, ActA. This fusion protein is accumulated on the outer membrane of mitochondria and binds covalently to a chloroalkane-tagged molecule in the cytosol. The cells were exposed to the chloroalkane-tagged peptides at 37 °C for 3 h. After washing, the cells were treated with a chloroalkane-tagged fluorescence dye, SiR-ct,

and analyzed by flow cytometry and confocal microscopy. The fluorescence intensity derived from the dye molecule decreased in a concentration-dependent manner and was well-fitted with sigmoidal curves (Fig. 2f). The sigmoidal fit showed that **DP1-L2Kct** translocated into the cytosol more effectively (CP₅₀ of 3.4 μM) than **CP1-L2Kct** (CP₅₀ of 7.4 μM) (Fig. 2e). The difference in permeability between **CP1** and **DP1** was not as large as the difference observed in the PAMPA and Caco-2 assay, which could be because the chloroalkane tag affects the permeability of the peptides and/or there is a difference among the permeation process across the cell membrane, artificial membrane, and cell monolayer. The result of the flow cytometry was confirmed by confocal microscopy (Fig. 2g and Supplementary Fig. 9), which revealed that red fluorescence from the cells decreased when incubated with **DP1-L2Kct**, but not significantly decreased when incubated with **CP1-L2Kct**. This cell-based permeability assay demonstrated that the cyclic depsipeptide **DP1** penetrates into the cytosol more effectively than the original cyclic peptide **CP1**.

## The effect of amide-to-ester substitution at an exposed amide NH on the stable conformations and lipophilicity of a cyclic hexapeptide

As discussed in the previous section, the higher membrane permeability of **DP1** and **DP5** than that of **CP1** is presumably because an exposed amide NH of **CP1** in the membrane is removed upon the amide-to-ester substitution. To evaluate the validity of this assumption, we conducted a conformational analysis of **DP1** in CDCl₃ which mimics the environment in a membrane. First, we reproduced the NMR structure of **CP1**. A similar conformation to that previously reported was obtained as the most stable conformation (Fig. 3a and Supplementary Figs. 10–15)[32]. In this conformation, amide hydrogen of Tyr-1 is exposed to solvent. Consistent with the previous report, the amide hydrogen of Leu-5 residue is not involved in intramolecular hydrogen bonding in the most stable conformation, but the amide hydrogen faces inward in the molecule and is probably partially masked from the solvent.

The conformational states of **DP1** in CDCl₃ were investigated using the same procedure as that of **CP1** (Fig. 3b and Supplementary Figs. 16–21). The superposition of **DP1** and **CP1** with their backbones showed that the most stable conformation of **DP1** is similar to that of **CP1** (Fig. 3c). The root-mean-square deviation (RMSD) of their backbones was calculated to be 0.426, which confirmed that the amide-to-ester substitution of the exposed amide NH does not significantly change the solution conformations of the cyclic hexapeptide in a membrane-like lipophilic environment. Therefore, the substitution reduces the total number of solvent-exposed amide hydrogens, leading to improved membrane permeability.

We next investigated the solution NMR structure of **MP1**. The RMSD between the backbones of **MP1** and **CP1** was 0.782. The value is small but higher than the RMSD between **DP1** and **CP1**. These results indicate that **MP1** forms a conformational state that is similar to but a little different from that of **CP1** in a lipophilic environment (Supplementary Figs. 22–28).

One of the possible reasons for the higher permeability of **DP1** than **MP1** can be the difference in the conformations between **DP1** and **MP1** in membrane. The NMR-based conformational analysis suggests that an amide-to-ester substitution does not largely change the conformation of **CP1** and removed a solvent-exposed amide NH of Tyr-1 residue while an amide *N*-methylation changed the conformation of **CP1** and did not reduce the total number of solvent-exposed amide NHs in a low dielectric environment. The conformational aspects were further assessed by amide temperature coefficient ($\Delta\delta$NH/$\Delta$T) measurements (Supplementary Table 3). While the trends in $\Delta\delta$NH/$\Delta$T values of Leu-2–Leu-5 residues of **CP1** and **DP1** are similar, that of **MP1** largely differ from those values of **CP1** and **DP1**. Of note, the $\Delta\delta$NH/$\Delta$T value of D-Leu-3 residue in **MP1** is smaller than −4.6 ppb/K, suggesting that the amide NH is exposed to solvent[13].

Another possible reason for the higher membrane permeability of **DP1** than **MP1** is that the lipophilicity of an ester bond is higher than that of an *N*-methylamide bond as suggested by the model dipeptide study. We calculated CLogP values to estimate the lipophilicity of these peptides. **MP1** has the highest CLogP (8.11), followed by **DP1** (7.52), and **CP1** (7.46). We also calculated ALogP values to estimate the lipophilicity of these peptides, which is a measure of compound lipophilicity calculated using a regression model based on the sum of the atomic lipophilicity of compounds. According to a previous report, ALogP value is a more accurate predictor of lipophilicity than CLogP for molecules with more than 45 atoms[38]. **DP1** has the highest ALogP (4.44), followed by **MP1** (4.00), and **CP1** (3.80). Since three-dimensional structures are not considered in these calculated lipophilicity values, the calculations suggest that the higher lipophilicity of an ester bond than that of an *N*-methylamide bond is one reason for the higher permeability of **DP1** than **MP1**.

Based on these results, the higher membrane permeability of **DP1** than **MP1** is assumed to be because **DP1** is more stable than **MP1** in a membrane-like low dielectric environment due to the smaller number of solvent-exposed amide NHs and the higher local lipophilicity of an ester bond. The experimentally determined 1,9-decadiene–water distribution coefficient log $D_\text{dec/w}$ was consistent with the assumption: **DP1** exhibited the highest log $D_\text{dec/w}$ (1.2), followed by **MP1** (0.052), and **CP1** (−0.74).

### Enhanced sampling simulations of membrane permeation of the cyclic hexapeptides CP1, DP1, and MP1 across lipid bilayer membrane

To understand how amide-to-ester substitution influences the dynamics of the conformational changes of the cyclic hexapeptide upon membrane permeation, we performed molecular dynamics (MD) simulations for reproducing the membrane permeation processes of **CP1**, **DP1**, and **MP1** across the lipid bilayer based on replica exchange with solute tempering/replica-exchange umbrella sampling (REST/REUS) method[39,40]. The simulations generated the conformational ensembles at each position along the reaction coordinate z, with 28 windows of harmonic potentials having different restraint centers. Simultaneously, the simulations were performed at 8 different effective temperatures of the solute for each restrained position. An MD simulation was performed for 300 ns after a 200 ns equilibration process for each peptide. The reaction coordinate z is defined as the distance on the axis orthogonal to the membrane plane between the center of mass of all the nitrogen atoms of phosphatidylcholine and that of the nitrogen atoms contained in the peptide bonds on the backbone of the peptides (Fig. 4a).

To reveal the conformational propensity at each position of the lipid bilayer, principal component analysis was performed using all trajectories, corresponding to the lowest temperature replicas of **CP1**, **DP1**, and **MP1**. In this analysis, 72-dimensional eigenvectors were calculated from the variance-covariance matrices constructed from 3D coordinates of the backbone atoms of residues 2–6, which are

common to **CP1**, **DP1**, and **MP1**. Then, the trajectories of **CP1**, **DP1**, and **MP1**, which were divided into three sections (inside (0–6 Å), at the interface of (7.5–22.5 Å), and outside the membrane (31.5–37.5 Å)) were projected against the first and second principal axes. The contribution from the first and second principal components was 51% and 14%, respectively. PC-1 seems to represent open and closed conformations. In this study, open conformations are defined as those with no intramolecular hydrogen bonds, while closed conformations are defined as those with three or more intramolecular hydrogen bonds. Inside the membrane (0–6 Å), all three peptides have a distribution of conformations concentrated in the same region, i.e., region A, in PCA space. A representative conformation of region A is extracted from the conformational ensemble of the **CP1** and named conformer A (Fig. 4b). Representative conformations of other regions are also named in the same manner. In aqueous solution (outside the membrane in Fig. 4b) (31.5–37.5 Å), the conformations of **CP1** and **DP1** are similar and concentrated in region C. The representative structure, i.e., conformer C, adopts an open conformation. Furthermore, **CP1** and **DP1** also adopt closed conformations in region A and conformations in-between open and closed in region D as represented by conformers A and D, respectively. On the other hand, the conformations of **MP1** are different from those of **CP1** and **DP1** in aqueous solution. The conformations are concentrated in region E, represented by an open conformer E. At the interface (z = 7.5–22.5 Å), the closed conformations in region A were found to be one of the major conformations for all three peptides (Fig. 4b). For **CP1** and **DP1**, open conformations in region C, conformations in region D, and closed conformations in region B were also found at the interface. The conformations in region B are unique ones only found at the interface. The conformations seem energetically favorable at the interface because the lipophilic moiety of the peptides can interact with the lipophilic lipid tail, and the hydrophilic moiety can interact with the hydrophilic lipid head and water. For **MP1**, the conformations in region E were found in addition to the closed conformations in region A at the interface. **DP1** formed conformations similar to **CP1** in the membrane, at the interface, and in water during the simulations. In contrast, **MP1** formed significantly different conformations from **CP1** at the interface and in water. The greater conformational difference between **MP1** and **CP1** compared with that between **DP1** and **CP1** is probably because amide *N*-methylation restricts the backbone bond rotations, and the conformational preference of **MP1** becomes different from **CP1** and **DP1**. The difference in the conformational preferences between **MP1** and **CP1/DP1** is reasonable considering the difference between the previously reported Ramachandran plots of model peptides with *N*-methylamides and those with amides/esters[18].

The conformational changes of the cyclic peptides during the process of membrane permeation suggested by the MD simulations were experimentally examined. In the membrane, the conformer A, which was the representative of the dominant conformational states, was similar to the conformations of **CP1**, **DP1**, and **MP1** determined by the NMR measurements in CDCl$_3$ (RMSD of their backbone was calculated to be 0.260, 0.433, and 0.908 Å for **CP1**, **DP1**, and **MP1**, respectively). In order to assess the validity of the conformations in an aqueous solution, NMR spectra of **CP1**, **DP1**, and **MP1** in a solvent with a high dielectric constant were measured (Supplementary Figs. 29–47). 1:1 mixture of DMSO and water was used as the solvent. DMSO was added to fully solubilize the peptides. The following two lines of evidence obtained from the NMR analysis suggested the validity of the conformations observed in an aqueous solution in the enhanced sampling MD simulations. First, for all three peptides, the number of inter-residue NOE signals was smaller in the high-dielectric solvent than in CDCl$_3$, suggesting that the peptides form more open conformations with a smaller number of intramolecular hydrogen bonds. Second, all the pairs of protons that gave NOE signals have average distances of less than 5 Å in the enhanced sampling MD simulations.

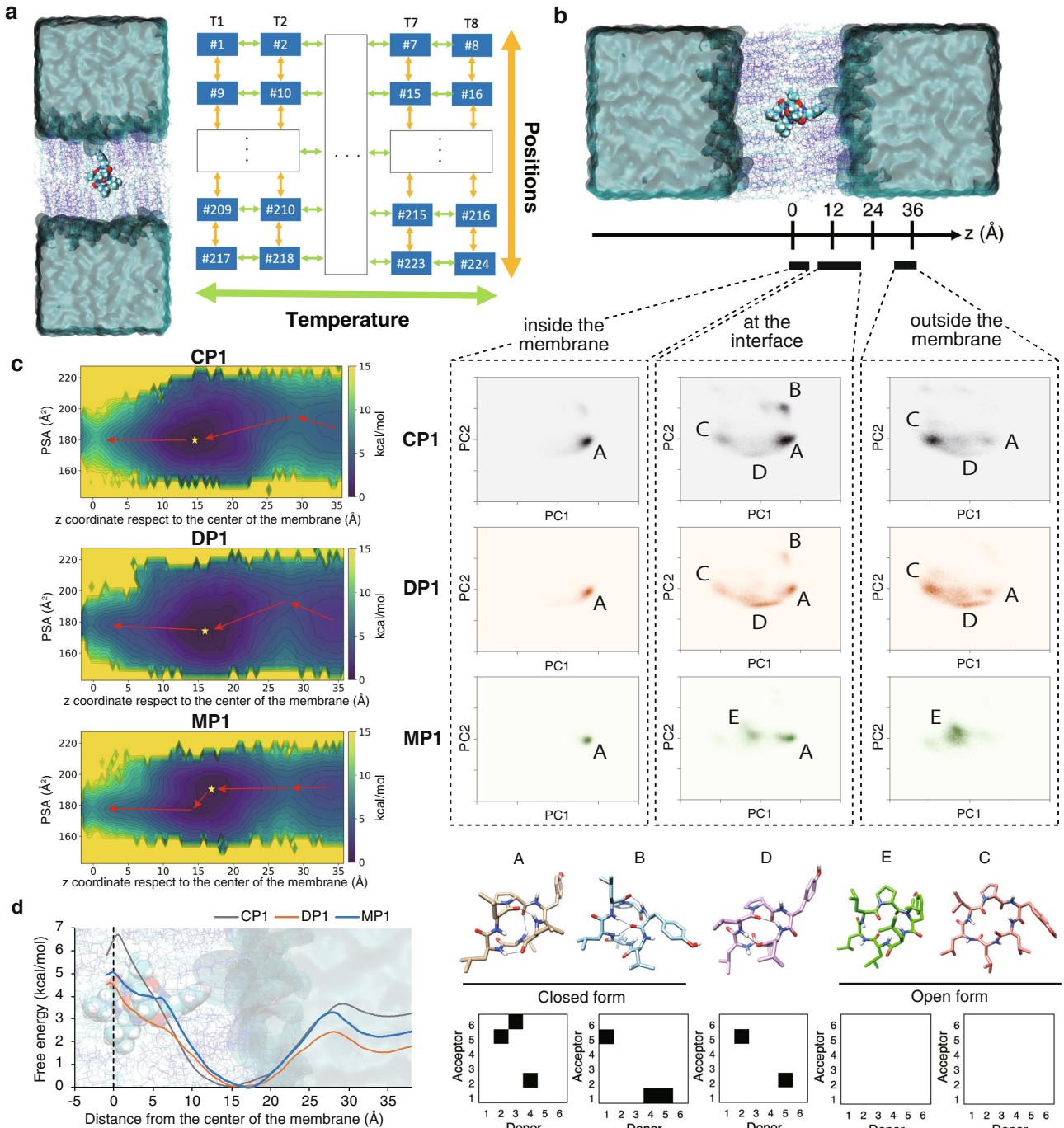

**Fig. 4 | Enhanced sampling MD simulations of the membrane permeation process of CP1, DP1, and MP1. a** Graphical abstract of the MD simulations. **b** Conformational ensembles of **CP1**, **DP1**, and **MP1** inside, at the interface, and outside the lipid membrane projected onto the first and second principal components (PC1 and PC2), and representative conformations and backbone hydrogen bonding patterns. The percentages of the major conformations are shown in Supplementary Fig. 48. **c** The two-dimensional free energy profiles against the polar surface area (PSA) and $z$ coordinate. Minima and saddle points in the free-energy profile are connected by red arrows. Stars denote the most stable positions. **d** The one-dimensional free energy profile of **CP1**, **DP1**, and **MP1** along the $z$ coordinate.

When the NMR-derived conformations and the representative conformations from the enhanced sampling MD simulations in the aqueous environment (conformer C for **CP1** and **DP1**, and conformer E for **MP1**) are compared (Supplementary Fig. 29), the RMSD value was 0.578 Å, 1.188 Å, 0.861 Å for **CP1**, **DP1**, and **MP1**, respectively, suggesting the consistency of the simulations and experiments. The RMSD is a little high for **DP1** and the conformation from the enhanced sampling MD simulations was more open than that from the NMR analysis. This can be explained by the fact that the simulations were conducted in water while NMR measurements were conducted in 1:1 mixture of

DMSO and water which has a smaller dielectric constant than that of pure water. Altogether, the NMR-based conformational analysis validates the conformational changes during the membrane permeation process suggested by the enhanced sampling MD simulations.

Next, to observe how the conformations of **CP1**, **DP1**, and **MP1** change during the entire membrane permeation process, two-dimensional free energy profiles against the polar surface area (PSA) and z coordinate were estimated based on weighted histogram analysis method (WHAM: Weighted Histogram Analysis Method., Version 2.0.9., http://membrane.urmc.rochester.edu/wordpress/?page_id=126)[41–43] and

depicted in Fig. 4c. For all the peptides, conformations with low PSA and high PSA, which, respectively, corresponds to closed and open conformers, were present in water and at the interface. Conformations with low PSA gradually became dominant as the peptides approached the center of the membrane. There is a notable difference in the PSA diagrams between **CP1/DP1** and **MP1**. While conformations with low PSA, corresponding to the conformations in region A or B in the PCA diagram, become the major population at the interface for **CP1** and **DP1**, conformations with high PSA, corresponding to those in region E in the PCA diagram, are dominant even up to $z = 17$ Å in the interface region for **MP1**. This reflects the difference in the conformational transitions during the permeation across the membrane between **CP1/DP1** and **MP1**, as shown with the PCA results (Fig. 4b).

For analyzing the energetic behavior, one-dimensional free energy profiles against the $z$ coordinate are depicted in Fig. 4d. There is a free energy minimum for all the peptides at the interface ($z = 15–17$ Å) and the barrier at the center of the membrane ($z = -0$ Å). A previous conformational study indicated that cyclic peptides with high membrane permeability have small energy gaps between the center of the membrane and the free energy minimum at the interface[44]. When the gaps are compared for the peptides tested in this study, **DP1** is the lowest, **MP1** is intermediate, and **CP1** is the highest. Consistent with the order of the free energy gaps, the efficiency of membrane permeation, as determined by PAMPA and Caco-2 assay, was higher for **DP1**, **MP1**, and **CP1** in that order. This consistency of the order of the free energy gaps in the simulations and the experimentally determined permeability indicates the validity of the simulations and that the difference of the membrane permeabilities among **DP1**, **MP1**, and **CP1** can be explained from the factors generating the difference of the energy gaps.

The MD simulations suggest that, during the membrane permeation process, **CP1**, **DP1**, and **MP1** dynamically change their conformations between the closed and open conformations in water and at the interface of the membrane. They need to adopt closed conformations in region A to be stable in a lipophilic environment when entering the membrane. **DP1** showed more similar behavior in conformational transition to **CP1** across membrane than **MP1**. For **DP1**, the number of exposed amide NHs of the closed conformer A is reduced compared with **CP1** by the amide-to-ester substitution, which in turn lowered the height of the free energy barrier at the center of the membrane relative to the minimum on the free energy profile. This reduction in free-energy cost for permeation is suggested to contribute to the higher membrane permeability of **DP1** than that of **CP1**. The degree of increase in membrane permeability of **MP1** compared with **CP1** was smaller than that of **DP1** in PAMPA and Caco-2 assay. Two possible reasons for this difference are suggested based on the simulations. First, **MP1** dominantly formed unique conformations, represented by conformer E, at the free energy minimum ($z = 17$ Å), which may increase the barrier for transition to conformations in region A, which are stable at the center of the membrane. Second, the relative stability of conformations in region A at the center of the membrane compared with the same conformations at the interface may be lower for **MP1** than that for **DP1**. As discussed with the NMR results, this is probably because the local lipophilicity of an $N$-methylamide is lower than that of an ester bond, and the number of solvent-exposed amide NHs is on average higher for **MP1** compared with **DP1**. These might lead to a higher energy gap from the interface to the center of the membrane for **MP1** than **DP1**.

## The effect of amide-to-ester substitution on membrane permeability of other cyclic peptides

To understand the scope and limitations of amide-to-ester substitution for improving membrane permeability of peptides, other cyclic peptides were also evaluated.

First, the membrane permeabilities of five diastereomers of **CP1** (**CP2–6**) were evaluated (Supplementary Fig. 49). These cyclic peptides were selected because the membrane permeability and conformational states have been analyzed in previous studies[45,46] (Supplementary Figs. 49a and b). These peptides (**CP2–6**) largely differ in permeability probably due to the difference in solvent-accessible surface area (SASA) in cyclohexane according to the previous reports. We introduced an amide-to-ester substitution on an exposed amide of the most stable conformation in cyclohexane solution to all the peptides. For all the five peptides, the amide NH of Tyr-6 is exposed in the most stable conformation in cyclohexane solution; therefore, we introduced the substitution on Tyr-6 (**CP2-6YE**, **CP3-6YE**, **CP4-6YE**, **CP5-6YE**, and **CP6-6YE**) (Supplementary Fig. 49c)[45]. The permeability of each peptide was measured by PAMPA. For **CP2**, **CP3**, and **CP4**, whose permeabilities are low, membrane permeability was significantly increased by an amide-to-ester substitution (**CP2-6YE**, **CP3-6YE**, and **CP4-6YE**) (Supplementary Fig. 49d). In contrast, for **CP5** and **CP6**, whose permeabilities are high, the amide-to-ester substitution did not improve membrane permeability (Supplementary Fig. 49d). This is probably because the peptides already have high lipophilicity, and the excess lipophilicity introduced by an amide-to-ester substitution lowered the membrane permeability[47]. Another possible reason is that the amide NH of the Tyr-1 residue of **CP5** and **CP6** is involved in intramolecular hydrogen bondings as suggested from previous conformational studies in CDCl₃[46] and the substitution caused conformational changes on the cyclic peptides. We also synthesized derivatives of **CP2–CP6** that have an amide $N$-methylation (**CP2-6YM**, **CP3-6YM**, **CP4-6YM**, **CP5-6YM**, and **CP6-6YM**) (Supplementary Fig. 49c) and evaluated the permeability of the peptides. As a result, in all the cases, the peptides showed lower membrane permeability than the corresponding peptides with an amide-to-ester substitution (Supplementary Fig. 49d). These results indicated that amide-to-ester substitution is an effective strategy for improving the membrane permeability of cyclic peptides with poor permeability.

We also examined the amide-to-ester substitution strategy for increasing the membrane permeability of cyclic peptides with a hydrophilic residue. **CP1** derivatives in which Tyr-1 residue is substituted with Phe residue and Leu-2 residue is substituted with Ser or Lys (**CP1-Y1F-L2S** and **CP1-Y1F-L2K**) (Fig. 5a). **CP1-Y1F-L2S** and **CP1-Y1F-L2K** have smaller ALogP values (1.94 and 1.75, respectively) than that of **CP1** (3.80), suggesting the lower lipophilicity of the two peptides compared with **CP1**. These two peptides were further modified at the Phe-1 residue with an amide-to-ester substitution or an amide $N$-methylation. The membrane permeability of these peptides was examined by PAMPA. The membrane permeability of **CP1-Y1F-L2S** was significantly increased by an amide-to-ester substitution while the permeability was not largely increased by an amide $N$-methylation (Fig. 5b left). The permeability value of **CP1-Y1F-L2K** was also increased by an amide-to-ester substitution (Fig. 5b right). Although the permeabilities of the original **CP1-Y1F-L2K** and the **CP1-Y1F-L2K** with an $N$-methylamide were under the quantification limits, the differences of the permeabilities with that of the **CP1-Y1F-L2K** with an ester were statistically significant considering the quantification limits ($7.0 \times 10^{-10}$ cm/s for the original **CP1-Y1F-L2K** and $7.5 \times 10^{-10}$ cm/s for the **CP1-Y1F-L2K** with an $N$-methylamide). However, the permeability value of **CP1-Y1F-L2K** with an ester is still low ($4.9 \times 10^{-9}$ cm/s), and therefore, further modifications, such as $N$-alkylation[29,48] are desirable for practical applications of peptides with charged residues like **CP1-Y1F-L2K**.

Next, we examined the effect of an amide-to-ester substitution on the membrane permeabilities of cyclic peptides with different ring sizes (Fig. 5c). **D8.31**, **D8.21**, and **D9.16** are cyclic 8- and 9-mer peptides with multiple $N$-methylated amides that are reported to have high membrane permeabilities[49]. We synthesized the peptides and their derivatives in which an $N$-methylated amide was substituted with a non-$N$-methylated amide (**D8.31-amide**, **D8.21-amide**, and **D9.16-**

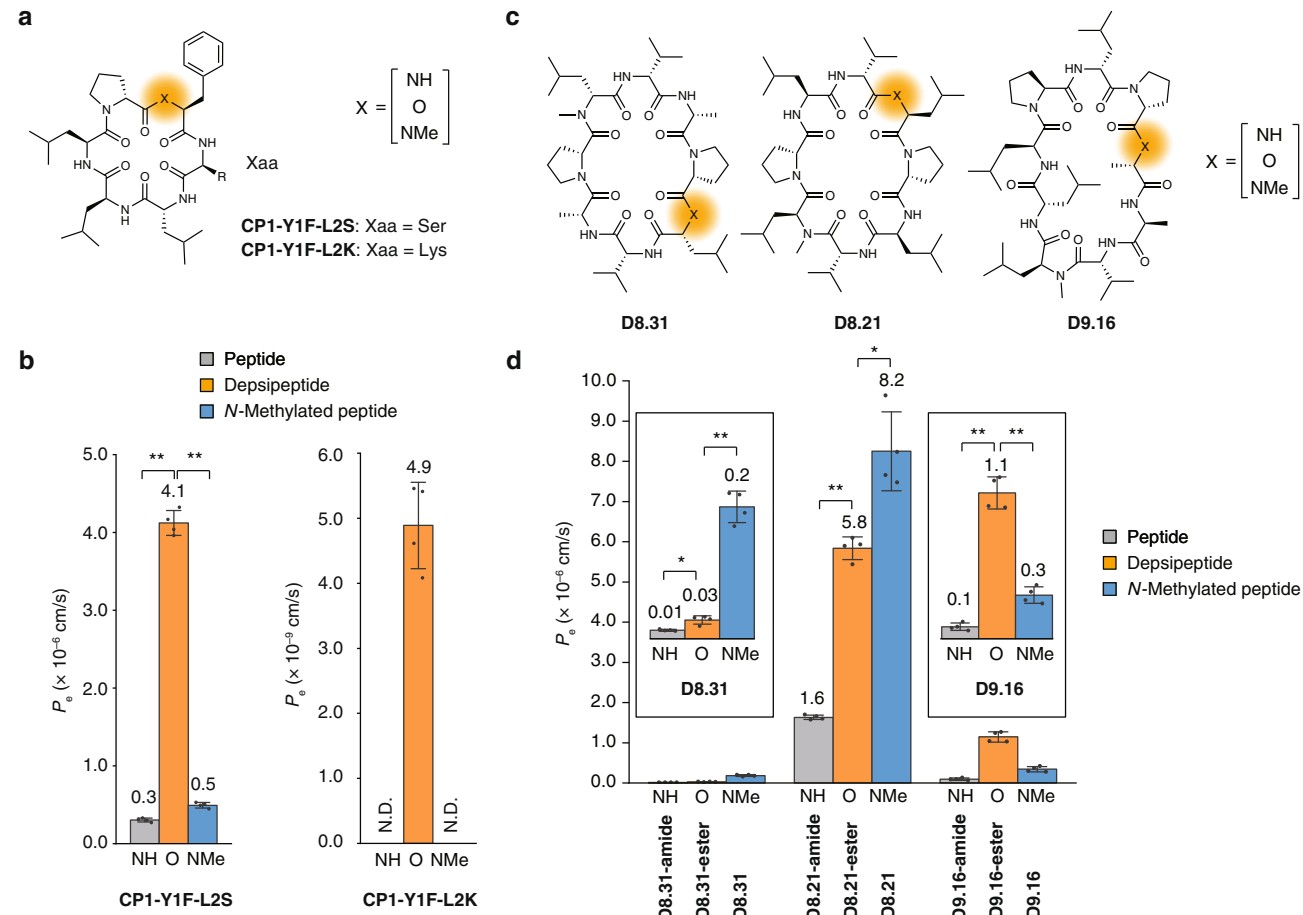

**Fig. 5 | The effect of an amide-to-ester substitution on cyclic hexapeptides with a hydrophilic residue and cyclic 8- and 9-mer peptides. a** The structures of **CP1** derivatives with a hydrophilic residue and their derivatives with an amide-to-ester substitution or an amide *N*-methylation. **b** The membrane permeabilities of the **CP1** derivatives shown in Fig. 5a. N.D. denotes not detected. *p* (**NH** vs. **O**) < 0.0001, *p* (**O** vs. **NMe**) < 0.0001 for **CP1-Y1F-L2S**. Note that the scale of the y-axis is different between the left and right graphs. **c** The structures of **D8.31**, **D8.21**, and **D9.16**, and their derivatives with substitution of an *N*-methylated amide with an amide (**D8.31-amide**, **D8.21-amide**, and **D9.16-amide**) or an ester (**D8.31-ester**, **D8.21-ester**, and **D9.16-ester**) and examined their permeabilities by PAMPA (Fig. 5d). In the reported conformations of **D8.31**, **D8.21**, and **D9.16** in CDCl$_3$, all the *N*-methyl groups are exposed to solvent; therefore, the removal of an *N*-methyl group from the peptides is expected to increase the number of solvent-exposed amide NHs, and **D8.31-amide**, **D8.21-amide**, and **D9.16-amide** show lower permeability while **D8.31-ester**, **D8.21-ester**, and **D9.16-ester** show higher permeability than **D8.31-amide**, **D8.21-amide**, and **D9.16-amide**. As expected, the original *N*-methylated series (**D8.31**, **D8.21**, and **D9.16**) and the depsipeptides (**D8.31-ester**, **D8.21-ester**, and **D9.16-ester**) showed higher permeability than the non-*N*-methylated peptides (**D8.31-amide**, **D8.21-amide**, and **D9.16-amide**). When the original *N*-methylated peptides and the derivatized depsipeptides are compared, the difference in permeability is sequence-dependent. For the two octapeptides (**D8.31** and **D8.21**), the ester versions showed lower permeabilities than the original *N*-methylated peptides while, for the nonapeptide (**D9.16**), the ester version showed higher permeability than the original *N*-methylated peptide. This is probably because an amide *N*-methylation and an amide-to-ester substitution differently affect the conformational transitions of the cyclic peptides during membrane permeation as demonstrated for **CP1**. These results showed that amide-to-ester

**D9.16-ester**). **d** PAMPA of cyclic 8-mer and 9-mer peptides. The enlarged views of the results of **D8.31** series and **D9.16** series are shown in the insets. PAMPA was conducted with 3 μM compounds in PBS containing 5% DMSO and 16 h incubation at 25 °C. Each bar represents the mean value, and the error bars the standard deviation from experiments carried out in quadruplicate. *p* (**D8.31-amide** vs. **D8.31-ester**) = 0.0138, *p* (**D8.31-ester** vs. **D8.31**) = 0.0004, *p* (**D8.21-amide** vs. **D8.21-ester**) <0.0001, *p* (**D8.21-ester** vs. **D8.21**) = 0.0126, *p* (**D9.16-amide** vs. **D9.16-ester**) = 0.0003, *p* (**D9.16-ester** vs. **D9.16**) = 0.0002. All the *P* values were determined by a two-sided Welch's *t*-test. \*\**p* < 0.01, \**p* < 0.05.

substitution is a useful choice for increasing membrane permeability of not only cyclic hexapeptides, but also larger cyclic peptides, although further studies on a more expanded set of large cyclic peptides are desirable in the future to understand the scope and limitation of the substitution.

## The effect of an amide-to-ester substitution on proteolytic stability and water solubility

Ester bonds are considered vulnerable to enzymatic degradation. Therefore, a plausible drawback of introducing amide-to-ester substitutions is that the substitution reduces the stability of peptides. To investigate whether this is the case, we measured the enzymatic stability of **DP1** in mouse serum (Fig. 6). The enzymatic stabilities of **CP1** and **MP1** were also measured for comparison. In addition, the stability of the corresponding linear depsipeptide (**DP1L-NH$_2$**) (Fig. 6a) was also measured to evaluate the effect of cyclization on enzymatic stability. All the cyclized peptides, namely **CP1**, **DP1**, and **MP1**, were stable in mouse serum for up to 24 h, while the linear depsipeptide **DP1L-NH$_2$** was degraded within 2 h (Fig. 6b). We also measured mouse plasma stability of **CP1**, **DP1**, **MP1**, and other cyclic depsipeptides and *N*-methylated peptides (**DP2–5** and **MP2–5**). All the cyclic depsipeptides and *N*-methylated peptides showed high stability in mouse plasma

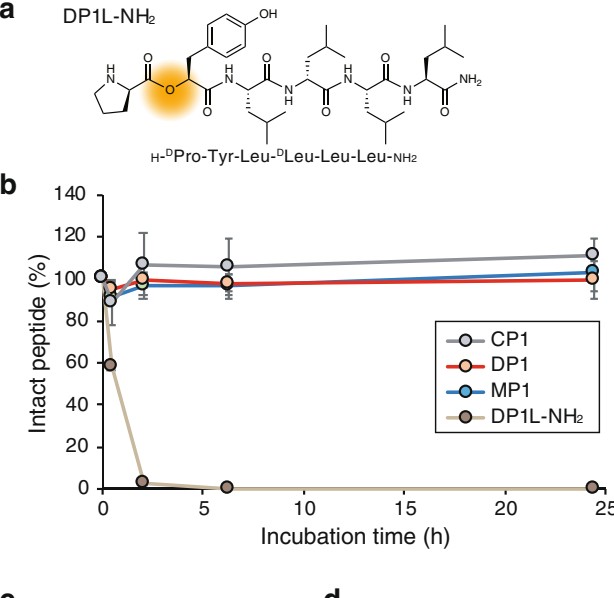

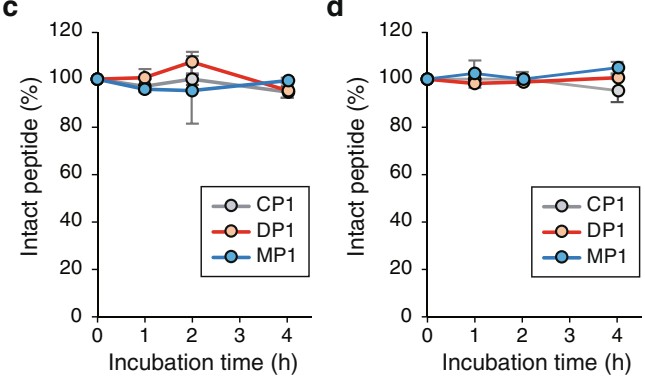

**Fig. 6 | Stabilities of cyclic hexapeptides in serum and simulated gastric/intestinal fluids. a** Chemical structure of **DP1L-NH₂**. **b** Mouse serum stability of **CP1** (gray), **DP1** (orange), **MP1** (blue), and **DP1L-NH₂** (brown). **c** Stability in a simulated gut fluid. 98% of a control peptide (somatostatin) was degraded at 4 h under the same conditions. **d** Stability in a simulated intestinal fluid. 94% of a control peptide (oxytocin) was degraded at 4 h under the same conditions. The degradation profiles of the control peptides are shown in Supplementary Fig. 50. In (**b**)–(**d**), each point represents the mean value, and the error bars the standard deviation from experiments carried out in triplicate.

(Supplementary Table 4). These results suggest that cyclic depsipeptides are as stable as cyclic peptides without ester bonds, and the macrocyclization probably contributes to the high stability. The effect of macrocyclization on shielding an ester bond from enzymatic degradation may depend on the sequence and the ring size of cyclic peptides. Such sequence/ring size dependence on the enzymatic stability is an interesting subject of a future study of cyclic depsipeptides.

Stability of **CP1**, **DP1**, and **MP1** in simulated gastric fluid (SGF) and simulated intestinal fluid (SIF) were also examined (Fig. 6c, d). Each peptide was incubated in SGF-containing pepsin or SIF-containing pancreatin for 1–4 h. As a result, no significant degradation of **CP1**, **DP1**, and **MP1** was observed while more than 90% of the control peptides (somatostatin for SGF and oxytocin for SIF) were degraded under the same conditions. These results suggest that **CP1** will be stable against enzymatic degradation until they reach the intestine to be absorbed when they are orally administered, and amide-to-ester substitution and amide *N*-methylation do not decrease the enzymatic stability. Considering that the SGF and SIF assays were conducted at pH 1.2 and pH 6.8, respectively, **CP1**,

**DP1**, and **MP1** are also suggested to be stable against hydrolysis at typical pH conditions in vivo.

The water solubilities of **CP1**, **DP1–5**, and **MP1–5** were measured in phosphate buffer (pH 7.4) containing 1.2% DMSO (Supplementary Table 4). The water solubility of **CP1** was greater than 100 µM under this condition. Compared with **CP1**, depsipeptides and *N*-methylated peptides showed lower water solubilities of 5–56 µM and 4–91 µM, respectively. However, these values are higher than or equivalent to the water solubility of an orally bioavailable peptide, cyclosporin A (6 µM). These results suggest that peptides with an amide-to-ester substitution can be useful for biological applications from the aspect of water solubility, though the substitution often lowers water solubility of the cyclic peptides.

## Discussion

In summary, we have shown that the backbone amide-to-ester substitutions increase the membrane permeability of dipeptides and cyclic hexapeptides. While the effect of introducing amide-to-ester substitutions on membrane permeability of small molecules and spacers of bivalent ligands has been evaluated in previous studies[50,51], this is the first study to directly evaluate the effect of introducing amide-to-ester substitutions on the membrane permeability of cyclic peptides. Cyclic peptides form complex three-dimensional structures that dynamically change in a short time scale. Therefore, understanding and controlling the membrane permeability of cyclic peptides present unique challenges.

The present study on the conformations and physicochemical properties of a model cyclic hexapeptide and its derivatives with an amide-to-ester substitution indicates the following two intriguing insights on features of amide-to-ester substitutions: (1) By substituting an amide bond whose hydrogen is exposed in a lipophilic environment to an ester bond, the membrane permeability of the cyclic peptide can be improved, (2) the amide-to-ester substitution of cyclic peptides does not decrease proteolytic stability as long as the substitution is introduced on cyclic hexapeptides. The NMR-based conformational analysis and enhanced sampling MD simulations suggest the dynamic conformational transition of the cyclic peptides among open and closed conformations upon permeation across lipid bilayer membrane. The conformational transition of **DP1** across the membrane is similar to that of the original peptide **CP1**. Interestingly, an amide-to-ester substitution increased the membrane permeability of the cyclic hexapeptides more than amide *N*-methylation in some of the tested cases.

In the present study, we have also examined the scope of an amide-to-ester substitution for increasing the membrane permeability of larger cyclic peptides. The result suggests that the substitution can increase the membrane permeability of cyclic 8- and 9-mer peptides although it can be sequence-dependent. Further studies on the effects of an amide-to-ester substitution on membrane permeability as well as conformational ensembles of large cyclic peptides are expected to facilitate more strategic applications of the substitution for increasing membrane permeability of cyclic peptides.

Currently, *N*-methylation of exposed amide NH groups is often used for improving the membrane permeability of peptides to reduce the energetic cost of amide NH dehydration by eliminating the amide NH groups. This study demonstrated that amide-to-ester substitution is another useful strategy for the same purpose. Our enhanced sampling MD simulations suggested that an amide-to-ester substitution and an amide *N*-methylation differently influence the conformational transitions during the membrane permeation process of a cyclic peptide and the difference in conformational transitions influences the membrane permeability. For some of the tested peptides in this study, an amide-to-ester substitution increased the membrane permeability more than an amide *N*-methylation while, for some other peptides, the result was the opposite. Therefore, we envision that the two backbone modifications will be utilized complementary for increasing membrane

permeability of cyclic peptides. In addition, we showed that cyclic depsipeptides are stable in serum/plasma at least for cyclic hexapeptides. Taken together, we envision that this amide-to-ester substitution strategy will be utilized for the development of peptides with high oral bioavailability, ability to target intracellular biomolecules, or both.

## Methods

### Synthesis
(S)-3-[4-tert-Butoxyphenyl]-2-hydroxypropionic acid, or an α-hydroxy acid derivative of Tyrosine with tBu protecting group was synthesized using a previously reported procedure for synthesizing other α-hydroxy acids via diazotization[52]. Peptides were synthesized by Fmoc solid-phase peptide synthesis. For cyclic peptides, precursor linear peptides were synthesized on 2-chlorotrityl chloride resin. After cleavage by 2,2,2-trifluoroethanol solution and purification by HPLC, the precursor peptides were cyclized using PyAOP, HOAt, and N,N-diisopropylethylamine (DIPEA) in solution. The synthesized cyclic peptides were purified by HPLC. Condensation of Fmoc amino acids or α-hydroxy acids with a terminal amine of the growing chain of the peptide was performed using HATU and DIPEA. Ester formation was performed using N,N-diisopropylcarbodiimide, and N,N-dimethylaminopyridine[53]. Amide N-methylation was performed using Fukuyama amine synthesis involving Mitsunobu reaction[54,55]. Details for synthesis are described in Supplementary Information.

### PAMPA
The permeability assay across the artificial membrane was conducted according to a previously reported procedure[56] with minor modifications. MultiScreen-IP Filter Plate, 0.45 μm (Merck) was loaded with 5 μL of 1% lecithin from soybean (Alfa Aesar) in dodecane. 300 μL of 5% DMSO/PBS was added to an acceptor well and 150 μL of peptide solution in 5% DMSO/PBS was added to a donor well. The donor plate was docked on the acceptor plate and the plates were incubated in a box containing wet paper towels at 25 °C. After incubation, the peptide concentrations in donor and acceptor wells were quantified by LC-MS. Multiple measurements were taken using samples from distinct wells. All the statistical tests are a two-sided Welch's t-test.

### Caco-2 assay
The permeability assay across cell monolayers was conducted according to a previously reported procedure[57] using Millicell cell culture insert plates (Millipore). $4.2 \times 10^4$ Caco-2 cells were spread on each insert chamber and culture medium was added in a receiver plate. The cells were cultured in the insert chambers at 37 °C for 22–25 days. The trans-epithelial electrical resistance (TEER) was measured to confirm the value in each well is above 300 Ω cm². The medium in the lower chamber and upper chamber was removed by aspiration. 0.4 mL of transport buffer (HBSS buffer containing 10 mM HEPES, pH 7.4) was added to the upper chamber and 0.8 mL of the same buffer was added to the lower chamber. After incubating the plate for 30 min at 37 °C, the buffer was removed. 0.42 mL of peptide solution in transport buffer was added to the upper chamber and 20 μL was aliquoted. 0.8 mL of transport buffer was added to the lower chamber. For evaluation in the basolateral-to-apical direction, 0.82 mL of peptide solution in transport buffer was added to the lower chamber and 20 μL was aliquoted. 0.4 mL of transport buffer was added to the upper chamber. The plate was incubated at 37 °C. 20 μL and 200 μL were aliquoted from the upper chamber and the lower chamber, respectively, and used for LC/MS analysis to determine the permeability coefficient. The solutions in both chambers were removed. 0.4 mL of 300 μM Lucifer yellow solution in transport buffer was added to the upper chamber and 0.8 mL of transport buffer was added to the lower chamber. After incubating the plate at 37 °C for 1 h, 200 μL of lower chamber solution was transferred to a 96-well black plate and the fluorescence at ex/em = 485/538 nm was measured. Multiple

measurements were taken using samples from distinct wells. All the statistical tests are two-sided Welch's t-tests.

### CAPA
The permeability assay using HeLa cells was conducted according to a previously reported procedure[36,37] with minor modifications as follows. HeLa cells expressing Haloenzyme-GFP with a mitochondria-targeting sequence were established as described in the Supplementary Methods. $4 \times 10^5$ cells were spread on a 96-well plate and cultured at 37 °C for 18 h. After the incubation, the cells were washed with PBS and incubated with compounds in RPMI containing 1% DMSO at 37 °C for 4 h. The cells were washed with PBS twice, and incubated with 1 μM HaloTag® SaraFluor™ 650 T Ligand (GORYO Chemical) in RPMI containing 0.1% DMSO at 37 °C for 30 min. The fluorescence dye solution was removed, and the cells were incubated with TrypLE Express (Thermo Fisher Scientific) at 37 °C for 10 min. After addition of RPMI containing 10% FBS and 1% Antibiotic-Antimycotic (Nakalai), the cells were collected in a test tube and washed with PBS. The cells were analyzed using a flow cytometer (Guava easyCyte, Merck Millipore, Massachusetts, USA). 3000 cells after gating by FSC/SSC were analyzed in each measurement. The presented data was derived from the mean fluorescence intensity derived from SiR-ct normalized by the fluorescence intensity from GFP. Representative histograms are shown in Supplementary Fig. 60.

### Confocal microscopy
$5.0 \times 10^4$ cells were spread on an 8-well cover grass chamber which was precoated with poly-L-Lysine and incubated for 14 h. The cells were washed with RPMI twice and incubated with compounds in RPMI containing 0.1% DMSO at 37 °C for 3 h. The cells were washed with RPMI twice and incubated with 1 μM HaloTag® SaraFluor™ 650T Ligand (referred to as SiR-ct) (GORYO Chemical) in RPMI containing 0.1% DMSO at 37 °C for 15 min. After being washed with RPMI twice, the cells were imaged using a Leica model TCS SP8 confocal laser-scanning microscope.

### NMR measurement
Lyophilized peptides were directly dissolved in a solvent for NMR measurements. NMR spectra were recorded on a Bruker AVANCE-III HD 800 spectrometer (Bruker, Billerica, MA, USA), equipped with a cryogenic probe at 298 K, unless otherwise stated. Two-dimensional TOCSY experiments were performed using standard pulse sequences and phase cycling (mlevetgp and mlevesgpph) with a mixing time of 60 ms, and EASY-ROESY and ROESY spectra were recorded using the previously reported pulse program and parameters[58] (roesyadjsphpr and roesyesgpph) with a mixing time of 250 ms. In the TOCSY, EASY-ROESY, and ROESY experiments, the spectral widths were set to 7211 or 8012 Hz for both dimensions, and 2048 × 1024 complex points were recorded. Two-dimensional CLIP-COSY experiments were performed using standard pulse sequences and phase cycling (clipcosy and clipcosyesgpph). The spectral widths were set to 7211 or 8012 Hz for both dimensions, and 2048 × 256 complex points were recorded. Two-dimensional $^1$H-$^{13}$C HSQC and HMBC experiments were performed using standard pulse sequences and phase cycling (hsqcetgpsp and hmbcgplpndqf). In the HSQC experiments, the spectral widths were set to 8012 Hz and 32,193 Hz for the $^1$H and $^{13}$C dimensions, respectively, and 1024 × 256 complex points were recorded. In the HMBC experiments, the spectral widths were set to 8012 Hz and 40,242 Hz for the $^1$H and $^{13}$C dimensions, respectively, and 2048 × 1024 complex points were recorded. The inter-scan delays were set to 1.3 s in all two-dimensional experiments. The temperature dependence of amide proton resonances was derived from $^1$H-1D spectra recorded on a Bruker Avance 600 spectrometer. Spectra were measured between 278 K to 303 K in 5 K increments and referenced to TMS at 0 ppm. All of

the spectra were processed and analyzed by the Topspin 3.5 or 4.1 software (Bruker).

## NMR structure calculations

The structures of **CP1**, **MP1**, **DP1**, and **DP2** were calculated by a simulated annealing protocol with the software XPLOR-NIH[59] using the distance restraints defined from the intensities of signals in their respective 2D ROESY NMR spectra. Topologies and parameters for the non-natural amino acids and the residue connections were generated by manual modification of those of tyrosine and peptide bond, respectively, using bonds, angles, and charge values generated by PRODRG[60]. The numbers of NOE distance restraints used for calculations were 86, 87, 80, and 92 for **CP1**, **MP1**, **DP1**, and **DP2** in chloroform, respectively, and 77, 72, and 68 for **CP1**, **MP1**, and **DP1** in DMSO/water, respectively. The NOEs were classified as strong, medium, weak, and very weak, and only the upper distance limits were set to allow greater conformational freedom. From 100 calculations, we selected the 10 lowest energy structures with no violations of the distance restraints >0.5 Å. NOE and J values used for the calculations are summarized in Supplementary Tables.

## MD simulations

The MD simulations were conducted according to our previously developed method[40]. The membrane permeation processes of target peptides were simulated based on the REST/REUS method[39]. The REST/REUS simulation was carried out with 224 replicas which consist of 28 windows with different restraint centers of a harmonic potential and 8 windows with different temperatures of the solute. The restraint center and force constant of the harmonic potentials were set to 1.0 Å interval and 1.5 kcal/mol/Å$^2$ for the section from $z = 0$ Å to 6 Å and 1.5 Å interval and 0.5 kcal/mol/Å$^2$ for the section from $z = 6.0$ to 37.5 Å. The temperature of peptides for each temperature replica is set to 300 K, 340 K, 390 K, 455 K, 540 K, 645 K, 785 K, and 980 K. The simulation consisted of 200 ns equilibration, followed by a 300 ns production run. Exchanges between adjacent replicas were attempted every 10 ps using the Metropolis scheme. The initial coordinates are extracted from the trajectory of pre-executed steered MD[61] with solute tempering[62]. The center of mass of the nitrogen atoms of peptide bonds was pulled from $z = 40.0$ Å, a position slightly beyond the reaction coordinate for the REST/REUS simulation, to −5.0 Å, a position slightly beyond the center of the membrane. A pulling rate of 0.25 Å/ns and force constant of 3.0 kcal/mol/Å$^2$ were used. In this process, the temperature of the peptide is set to 2100 K to obtain diverse conformations. All MD simulations were performed using the GPU-accelerated PMEMD module (pmemd.cuda) of the AMBER 20 software package (Case D. A. et al. AMBER 2020, University of California, San Francisco). Peptide and POPC molecules are parameterized with Amber10:EHT parameter set in the MOE (Molecular Operating Environment, 2019.01; Chemical Computing Group ULC, 1010 Sherbrooke St. West, Suite #910, Montreal, QC, Canada, H3A 2R7.) and Lipid 17 force fields (Gould I. R. et al., Lipid17: A comprehensive AMBER force field for the simulation of zwitterionic and anionic lipids. *in preparation*). The TIP3P model is employed as water molecules[63]. Peptide conformations were analyzed by principal component analysis based on the 3D coordinates of the backbone atoms of residues 2–6 (24 atoms), the common chemical structure of **CP1**, **DP1**, and **MP1**. 72-dimensional eigenvectors were obtained by diagonalizing the variance-covariance matrix calculated from all snapshots of the three peptides (630,000 snapshots) with the temperature of 300 K superimposed on the 2–6 residues of the backbone atoms. Subsequently, trajectories corresponding to the inside, at the interface, and outside the lipid membrane of each peptide were projected onto the eigenvectors corresponding to the first and second principal components. After

that, the representative conformations, labeled as A to E, were extracted from the peaks of the distributions. Then, the percentage of all conformations in a rectangular region in the two-dimensional PCA space around the representative conformation is estimated (Supplementary Fig. 48). The rectangular region in PCA space containing conformations A–D was defined as the region containing conformations within 0.4 Å of the RMSD of the backbone atoms of residues 2–6 from the representative structure in the trajectory of the interface of **DP1**. In calculating the percentage of conformations A–D, the rectangular region on the PCA space defined here was adopted for all peptides. The rectangular region in PCA space containing conformations E was defined as the region containing conformations within 0.4 Å of the RMSD of the backbone atoms of residues 2–6 from the representative structure in the trajectory of the outside of the membrane of **MP1**.

## Mouse serum and plasma stability

For **CP1**, **DP1**, its linear precursor, and **MP1**, the time course of degradation in mouse serum was evaluated. Compounds were incubated in mouse serum containing 1% DMSO at 37 °C for 24 h in triplicate. At each time point, 20 μL of the solution was aliquoted and diluted four times using acetonitrile to precipitate serum proteins. After centrifugation, the supernatant was diluted with water and the concentration of the intact peptide was quantified by LC-MS. For other peptides, the amount of residual intact peptides was evaluated after incubation in mouse plasma at 37 °C for 30 min. The amount of intact peptides was determined by taking the ratio of the LC-MS peak area of intact peptides just before and after the incubation.

## Solubility assay

Compounds were incubated in 0.1 M phosphate buffer (pH 7.4) containing 1.2% DMSO at 37 °C for 4 h. After the incubation, the solution was filtered, and the filtrate was analyzed by LC-UV (274 nm) to quantify the concentration of dissolved compounds.

## Stability assays in simulated gastric fluid (SGF)

The stability assay of peptides in SGF was conducted according to United States Pharmacopeia & National Formulary (USP 25-NF 20)[64] and the previous report[65] with some modifications as follows. SGF used in this study was prepared by dissolving pepsin (3.6 mg/mL) (Sigma, P7012, 3357 Unit/mg) in an aqueous solution of 2 mg/mL sodium chloride (NaCl) at pH 1.2. The solution was vortexed for 1 min, sonicated for 15 min, centrifuged at 20,000 × $g$ for 10 min, and filtered before use. 57 μL of SGF was incubated at 37 °C for 15 min and 3 μL of 1 μM peptide solution in DMSO was added to the warmed SGF. After 1, 2, or 4 h, 60 μL of ice-cold ultra-pure water containing 5% trifluoroacetic acid was added and 120 μL of 30% acetonitrile in ultra-pure water was added. The samples were filtered and injected into UPLC.

## Stability assays in simulated intestinal fluid (SIF)

The stability assay of peptides in SIF was conducted according to United States Pharmacopeia & National Formulary (USP 25-NF 20)[64] and the previous report[65] with some modifications as follows. SIF used in this study was prepared by dissolving pancreatin (1.25 mg/mL) (Sigma, P7545, 8 × USP specifications) in 50 mM phosphate buffer at pH 6.8. The solution was vortexed for 1 min, sonicated for 15 min, centrifuged at 20,000 × $g$ for 10 min, and filtered before use. 57 μL of SIF was incubated at 37 °C for 15 min and 3 μL of 1 μM peptide solution in DMSO was added to the warmed SGF. After 1, 2, or 4 h, 60 μL of ice-cold ultra-pure water containing 5% trifluoroacetic acid was added and 120 μL of 30% acetonitrile in ultra-pure water was added. The samples were filtered and injected into UPLC.

**Reporting summary**

Further information on research design is available in the Nature Portfolio Reporting Summary linked to this article.

## Data availability

The authors declare that the data supporting the findings of this study are available within the article and its supplementary information files. The raw data for assays, measurements, and simulations are available from the corresponding authors upon request.

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

## Acknowledgements

S.S. acknowledges financial support from CREST (JPMJCR21N5), Japan Science and Technology Agency. J.M. acknowledges financial support from PRESTO (JPMJPR21AF). Y.H. acknowledges financial support from KAKENHI (JP21J13541) and the Program for Leading Graduate School (MERIT). C.K., C.T., and S.L. acknowledge financial support from the US National Institutes of Health (GM131135). T.U. acknowledges financial support from MEXT/JSPS KAKENHI (JP21H05509, JP20H03375, and JP17H06097). K.T. acknowledges financial support from JSPS KAKENHI (JP20K21494, JP20H03378, and JP22K18374). M.Sugita and Y.A. acknowledge financial support from MEXT (Program for Building Regional Innovation Ecosystems) and KAKENHI (JP17H01814). This work was partly supported by Research Support Project for Life Science and Drug Discovery (Basis for Supporting Innovative Drug Discovery and Life Science Research (BINDS)) from AMED under grant numbers JP22ama121053. The NMR experiments were performed at NMR Platform of The University of Tokyo supported by MEXT, Japan. We thank One-stop Sharing Facility Center for Future Drug Discoveries at the University of Tokyo for the use of micrOTOF II.

## Author contributions

Y.H., J.M., and S.S. conceived the concept and designed the experi-ments. S.L., Y.A., J.M., and S.S. supervised the project. Y.H. performed the majority of the experiments and analyzed data. S.U. synthesized cyclic peptides and conducted PAMPA. M.Shinkai synthesized cyclic peptides and conducted a part of the NMR analysis. C.T., C.K., M.N., H.L., and S.L. performed the PAMPA and conformational analysis using NMR. K.K. and M.I. performed the PAMPA, Caco-2 assay, stability assay, and solubility assay. R.U. designed and supervised cell-based assays. T.U. and K.T. conducted NMR measurements and conformational analysis. M.Sugita and Y.A. performed the MD simulations. Y.H., J.M., and S.S. wrote the manuscript with contributions from all authors.

## Competing interests

The authors declare no competing interests.
