## [Peer Review File · Nature Communications]

Amide-to-ester substitution as a stable alternative to N-methylation for increasing membrane permeability in cyclic peptidesREVIEWER COMMENTS

Reviewer #1 (Remarks to the Author):

The manuscript submitted by Lokey and Sando et al. investigate the role of amide to ester substitution in hydrophobic dipeptides and cyclic hexapeptides as an alternative to N-methylation in increasing membrane permeability. Ester linkage within a cyclic peptide backbone is quite common in naturally occurring cyclic peptides possessing biological activities mediated through intracellular target binding and have been actively pursued as therapeutics, e.g., Echinomycin, Romidepsin, etc. Thus, cyclic depsipeptides are certainly worth investigating, particularly with special emphasis on their pharmacological properties.

N-methylation of cyclic peptides to lower their desolvation penalty for efficient transport across the lipid bilayer has been the most sought-after strategy to increase membrane permeability of cyclic peptides. However, due to unpredictable conformational changes, N-methylation can lead to the loss of biological activity of cyclic peptides. Thus, alternative strategies are desperately required to improve the membrane permeability of cyclic peptides.

This manuscript systematically evaluates the role of an ester bond in improving the membrane permeability of simple dipeptides and cyclic peptide model systems and thus would find utility beyond this report.

Nevertheless, I have several queries that the authors might clarify:

1. One of the significant findings emerging from the MD simulation is the dynamics between the open and closed conformation at the various region of the membrane that drives the permeation across the membrane. I wonder if the authors could experimentally evaluate the validity of this claim through NMR experiments, possibly in solvents with different dielectrics, and assess the exchange of the amide protons. Moreover, does the conformation of CP1, DP1, and MP1 differ in the high dielectric solvent?
2. The claim about the stability of depsipeptides against peptides needs rigorous validation. The authors have performed the stability studies in serum; however, it would be worth assessing the stability of the peptides in plasma. In an in vivo system, the peptides would interact with the plasma proteins. It is worth evaluating whether binding to the plasma proteins alters the stability of depsi vs. regular peptides. Additionally, since one of the primary goals is to derive oral availability for cyclic peptides, it would be interesting to see the stability of these peptides in gastric and intestinal fluids.
3. The authors discuss the enhanced permeability of DP2 at higher concentrations due to the involvement of efflux transporters. This should be evaluated by using efflux pump inhibitors.
4. It would be worth including a positive control to obtain a feel of the absolute permeability enhancement in the permeability plots.
5. I wonder why the PAMPA assay was done for 18 h, as opposed to 4 h. Does it relate to the kinetics of permeation? Also, can the authors explain the rationale behind the use of various PAMPA assay plates for different libraries?

Jayanta Chatterjee

Reviewer #2 (Remarks to the Author):

The authors describe a systematic study of the effect of amide-to-ester substitution on the permeability of a series of dipeptides and cyclic hexapeptides. They compare the results with those of the N-methylated analogues. The findings are overall very interesting and highly relevant to the field.

The manuscript is also well written. I have some comments for consideration before publication (in the order of appearance in the manuscript).

- While some of the findings should be applicable to other sizes and scaffolds of cyclic peptides, others might be less transferable. Therefore, I would find it important that already the abstract mentions the size of the cyclic peptides in the study (i.e. hexapeptides).
- Introduction: It is mentioned that there is a prevalence of depsipeptides with high permeability (line 80). I think it should also be noted that there are naturally occurring depsipeptides with low permeabilities (e.g. PF1022A, see <https://doi.org/10.1039/D0OB01447H>) and what possible reasons could be for the differences.
- Introduction, line 83: An example for a computational study on the amide-to-ester substitution in protein folding is <https://doi.org/10.1016/j.bbagen.2014.09.014>
- Lines 136-139: The NMR experiments of DP2 were again performed in chloroform? Was there any attempt to resolve the multiple conformers of DP2? Given that this peptide might adopt a different permeable conformation than CP1/DP1, it would be useful to determine it.
- Line 146/147: I think the second part of the sentence "DP2 did not show a significant improvement in permeability compared to CP1, as observed in PAMPA (Figure 2d)" is not correct. DP2 shows a higher permeability than CP1 in Figure 2c (as discussed on line 123-125).
- NOEs-MM calculations: Where the NOE restraints applied in a (time/ensemble)-averaged manner? It is not clear to me from the Methods section. The use of instantaneous restraints (i.e. each conformer is expected to fulfil the restraints) is problematic because the NOE-derived distances represent averaged values.
- MD simulations: It would be helpful if the length of the individual trajectories is also mentioned in the main text (or at least the caption of Figure 4) as this is an important information
- What were the input features/dimensions for the PCA analysis? This information is missing, also from the Methods section.
- Figure 4c does not really allow a quantitative comparison of the populations of the different regions. It looks like there is enough simulation data available to construct Markov state models such that the steady-state populations can be estimated. I think this is important for the interpretation of the relative conformational behaviour.
- The authors highlight in the discussion on page 13 (and also in the abstract and discussion on page 17) that "DP1 showed more similar behaviour in conformational transition to CP1 across membrane than MP1." While this is clearly the case, I do not entirely agree with the discussion on how this is connected to the permeability. CP1 is actually the least permeable of the three, so why should a similar conformational behaviour be beneficial for permeability? That CP1 and DP1 have the same conformational behaviour but different permeability suggests to me that the polarity/lipophilicity is the only relevant factor in the case of these cyclic hexapeptides, reducing the free-energy barrier between the interface and the membrane interior. Looking at Figure 4d and 4c, the same holds true also for MP1, because region A with the permeable conformation is actually significantly populated by all three peptides. The author may want to refine the discussion on page 13 taking these considerations into account.
- Other cyclic peptides, page 15: The results presented here are a bit puzzling to me. The authors write that "all five peptides have only one exposed amide bond (Tyr-6) in the most stable conformation in cyclohexane solution". However, despite this two of the peptides (CP5 and CP6) have high permeabilities while CP2-CP4 do not. Has this been investigated in the previous studies (Ref. 42,43)? Is there a conformational explanation for these differences? If yes, it would be helpful to summarise them here because this is likely important to understand better the varying effect of the amide-to-ester substitution.
- Proteolytic stability, page 16: Could the authors comment on the (expected) ring-size dependence of the proteolytic stability? While the conformation of the cyclic hexapeptides are compact enough that one can imagine that it is difficult for an enzyme to get access to the ester bond, this "shielding" will likely decrease with increasing ring size. I think this is an important aspect that should be discussed in this section.
- Overall: I would caution a bit regarding the generality of the applicability of the amide-to-ester substitution. The example of DP2 shows clearly that unexpected conformational changes can occur, impacting the permeability either positively or negatively. In addition, one can expect that the

observed proteolytic stability depends on the ring size of the cyclic peptides.

- Methods section, page 21: The details for the steered MD and the PCA analysis should be included. I think the sentence "The membrane permeability of the peptide is estimated based on the slightly modified inhomogeneous solubility-diffusion model (ISDM)" can be removed, because I haven't seen any such results in the main text.

Reviewer #3 (Remarks to the Author):

General Comments

This paper explores the effect of replacing amide and n-methylated amides from a highly studied and reported cyclic hexapeptide system with esters (depsipeptides) on membrane and cell permeability. The authors show that in some cases this replacement resulted in improved permeability across both artificial membranes and cell monolayers. The authors claim that the depsipeptides do not affect the three-dimensional structures of these peptides as much as n-methylation.

Whilst this work is of some limited interest in the field, the scope of what is presented is restricted to largely one cyclic hexapeptide system and some of these authors (and others) have published on amide to ester substitutions in small molecule systems that improve permeability, hence limiting the novelty of this work. <https://pubs.acs.org/doi/10.1021/acs.jmedchem.1c01496>

I don't consider that paper would appeal to a broad scientific audience and is better suited to a more specialized journal. However, to do that the authors still need to do a lot of work to correct some major flaws and provide missing support for suggested findings.

Specific Comments

Section 1- Dipeptides

The Authors state that for all sequences the depsipeptides exhibited the highest permeability, but Figure 1c shows that this is only clearly the case for D1. No statistical analysis of D2 vs M2 or D3 vs M3 is provided, and there does not appear to be much difference in these numbers. Importantly, these compounds have only very low permeability. How confident are the authors of the accuracy of the quantification of such small amounts of these peptides in the acceptor/donor wells?

The scope of this experiment is very limited. Why did the authors only choose to look at dipeptides? Other physical properties of those dipeptides such as LogP, HPLC elution times, LogD and solvation energy could also be analyzed.

Section 2 -Cyclic Peptides

Figure 2c and 2d- Again no statistical analysis is provided. Are all 5 comparisons statistically significant?

Figure 2f- No error bars at all are on this curve CP50 3.4 vs 7.4 μ M are reported with no error of measurement or statistical significance. Are these numbers actually different?

Figure 2g- The confocal microscopy appears to show a much larger difference than the flow cytometry results, which are virtually identical. Why is this the case?

The explanation of structural heterogeneity for DP2 (from line 132 to line 142) by NMR spectroscopy is inconclusive and confusing. Multiple sets of NMR signals for Tyr-1 could be due to impurity. If multiple sets of NMR signals are the result of exchangeable structures of DP2 in solution, then variable temperature and/or solution NMR experiments should be carried out to experimentally

support this claim.

Another confusing statement from the authors is that there are three sets of NMR resonances for Tyr-1 H α and H β , but only two sets of peaks for H α (lines 137 to 138). The H α and H α of Tyr-1 should have the same number of sets of signals.

There are inconsistent results for CP1 cell permeability as figure 2c and 2d indicated CP1 has very low membrane and cell permeability. However figure 2e, 2f and 2g showed CP1 with chloroalkane tag can significantly penetrate through the cell membrane, is this result due to the tag itself entering the cell or enhancing permeability?

Section 3-

There are no NOE summaries, amide coupling constants, variable temperature amide chemical shifts included for solving 3D structures in the SI. These data are absolutely essential for investigating and verifying any solution structure (e.g. CP1, DP1 and MP1), and for reviewing this paper.

Figure S11. ROESY spectrum for CP1 does not show the crucial inter-residue ROEs

Figure S17. ROESY spectrum for DP1 shows overlapping alpha proton signals for residues I3, L4 and p6. There was no discussion of how the authors derived ROEs from those overlapping signals. This ROESY spectrum also does not show any inter-residue ROEs, suggesting that the structure derived from it is unreliable.

Only AlogP was used to calculate predicted 'lipophilicity' - why not any other calculations? Do other calculations lead to the same predictions for amide to ester replacements? AlogP often does not accurately predict experimental logP data.

Section 4-

The in silico simulation is the strongest part of this work. This is why this work may be better suited for a computational chemistry journal.

It is crucial that modeling results in this section are supported by at least some experimental data (e.g. CD, NMR amide proton temperature dependence, amide coupling constants to name just a few). Modeling predictions are notoriously prone to erroneous interpretations when not supported by strong experimental evidence of structure to prove the model.

Section 5-

I find the title of this section misleading as only cyclic hexapeptides with the sequence [PLLLLY] were used. As such the authors' claims of being able to improve permeability in a range of peptides is misleading and actually pretty limited. No n-methyl comparisons were used in this section.

This paper would be greatly enhanced if ester replacements were made to a broader range of peptides and the effects on permeability and structure explored.

Again no statistical analysis.

Section 6-

Stability of peptides in different pH buffers should also be examined, since depsipeptides are well known to be prone to hydrolysis. Indeed, that is a major reason why depsipeptides have not translated well in medicinal chemistry - in most cases they tend to be quite unstable in vivo and prone to cleavage.

Discussion

Figure 6 - this figure is an oversimplification, the effect will not be just because of one water interaction.

Methods

PAMPA assay: Different assay systems were used for CP1 and derivatives than for the other peptides. Why?

The Genetest Pre-coated PAMPA plate can give very different results than the homemade system. All compounds need to be measured in the same system to make valid comparisons.

NMR measurements: Spectra are recorded in CDCl₃ but DMSO appears in some of the spectra. Does the presence of this more polar solvent affect the structures presented?

Authors: Why so many corresponding authors?

Reviewer 1

Original comments from Reviewer 1

The manuscript submitted by Lokey and Sando et al. investigate the role of amide to ester substitution in hydrophobic dipeptides and cyclic hexapeptides as an alternative to N-methylation in increasing membrane permeability. Ester linkage within a cyclic peptide backbone is quite common in naturally occurring cyclic peptides possessing biological activities mediated through intracellular target binding and have been actively pursued as therapeutics, e.g., Echinomycin, Romidepsin, etc. Thus, cyclic depsipeptides are certainly worth investigating, particularly with special emphasis on their pharmacological properties.

N-methylation of cyclic peptides to lower their desolvation penalty for efficient transport across the lipid bilayer has been the most sought-after strategy to increase membrane permeability of cyclic peptides. However, due to unpredictable conformational changes, N-methylation can lead to the loss of biological activity of cyclic peptides. Thus, alternative strategies are desperately required to improve the membrane permeability of cyclic peptides.

This manuscript systematically evaluates the role of an ester bond in improving the membrane permeability of simple dipeptides and cyclic peptide model systems and thus would find utility beyond this report.

Nevertheless, I have several queries that the authors might clarify:

1. One of the significant findings emerging from the MD simulation is the dynamics between the open and closed conformation at the various region of the membrane that drives the permeation across the membrane. I wonder if the authors could experimentally evaluate the validity of this claim through NMR experiments, possibly in solvents with different dielectrics, and assess the exchange of the amide protons. Moreover, does the conformation of CP1, DP1, and MP1 differ in the high dielectric solvent?
2. The claim about the stability of depsipeptides against peptides needs rigorous validation. The authors have performed the stability studies in serum; however, it would be worth assessing the stability of the peptides in plasma. In an in vivo system, the peptides would interact with the plasma proteins. It is worth evaluating whether binding to the plasma proteins alters the stability of depsi vs. regular peptides. Additionally, since one of the primary goals is to derive oral availability for cyclic peptides, it would be interesting to see the stability of these peptides in gastric and intestinal fluids.

3. The authors discuss the enhanced permeability of DP2 at higher concentrations due to the involvement of efflux transporters. This should be evaluated by using efflux pump inhibitors.
4. It would be worth including a positive control to obtain a feel of the absolute permeability enhancement in the permeability plots.
5. I wonder why the PAMPA assay was done for 18 h, as opposed to 4 h. Does it relate to the kinetics of permeation? Also, can the authors explain the rationale behind the use of various PAMPA assay plates for different libraries?

Point-by-point response to the comments of Reviewer 1

We thank for the constructive suggestions. We provide answers to all the reviewer's comments as follows.

Comment from the reviewer

1. One of the significant findings emerging from the MD simulation is the dynamics between the open and closed conformation at the various region of the membrane that drives the permeation across the membrane. I wonder if the authors could experimentally evaluate the validity of this claim through NMR experiments, possibly in solvents with different dielectrics, and assess the exchange of the amide protons. Moreover, does the conformation of CP1, DP1, and MP1 differ in the high dielectric solvent?

>Our response to the comment

We thank the reviewer's critical comment on the results of the MD simulations.

To assess the validity of the MD simulations, we conducted NMR measurements of **CP1**, **DP1**, and **MP1** in a solvent with high dielectric constant to mimic the environment of MD simulations in water. Due to the poor solubility of the peptides in pure water, 1:1 = water:DMSO was used for the measurement.

The obtained data about the peptide conformations are consistent with the enhanced sampling MD simulations. This result validates the conformational changes observed in the MD simulations.

The following sentences describing the experimental results and discussion about the consistency with the MD simulations have been added to the manuscript.

[Manuscript p.20]

The conformational changes of the cyclic peptides during the process of membrane permeation suggested from the MD simulations were experimentally examined. In the membrane, the conformer A, which was the representative of the dominant conformational states, was similar to the conformations of **CP1**, **DP1**, and **MP1** determined by the NMR measurements in CDCl_3 (RMSD of their backbone was calculated to be 0.260, 0.433, and 0.908 Å for **CP1**, **DP1**, and **MP1**, respectively). In order to assess the validity of the conformations in an aqueous solution, NMR spectra of **CP1**, **DP1**, and **MP1** in a solvent with a high dielectric constant were measured (Figure S28–46). 1:1 mixture of DMSO and water was used as the solvent. DMSO was added to fully solubilize the peptides. The following two lines of evidence obtained from the NMR analysis suggested the validity of the conformations observed in an aqueous solution in the enhanced sampling MD simulations. First, for all three peptides, the number of inter-residue NOE signals was smaller in the high-dielectric solvent than in CDCl_3 , suggesting that the peptides form more “open” conformations with a smaller number of intramolecular hydrogen bonds. Second, all the pairs of protons that gave NOE signals have average distances of less than 5 Å in the enhanced sampling MD simulations. When the NMR-derived conformations and the representative conformations from the enhanced sampling MD simulations in the aqueous environment (conformer C for **CP1** and **DP1**, and conformer E for **MP1**) are compared (Figure S28), the RMSD value was 0.578 Å, 1.188 Å, 0.861 Å for **CP1**, **DP1**, and **MP1**, respectively, suggesting the consistency of the simulations and experiments. The RMSD is a little high for **DP1** and the conformation from the ~~steered~~ enhanced sampling MD simulations was more open than that from the NMR analysis. This can be explained by the fact that the simulations were conducted in water while NMR measurements were conducted in 1:1 mixture of DMSO and water which has a smaller dielectric constant than that of pure water. Altogether, the NMR-based conformational analysis validates the conformational changes during the membrane permeation process suggested by the enhance sampling MD simulations.

Figure S28. Comparison of the representative conformations in high dielectric solvents from NMR and enhance sampling MD simulations.

The overlay of the representative conformations of (a) **CP1**, (b) **DP1**, (c) **MP1** from NMR in 1:1 mixture of DMSO and water (green) and simulations in water (cyan). Two structures were overlaid using backbone atoms (N, C_α, C', and O).

Comment from the reviewer

- The claim about the stability of depsipeptides against peptides needs rigorous validation. The authors have performed the stability studies in serum; however, it would be worth assessing the stability of the peptides in plasma. In an in vivo system, the peptides would interact with the plasma proteins. It is worth evaluating whether binding to the plasma proteins alters the stability of depsi vs. regular peptides.

>Our response to the comment

The stability studies of the peptides in plasma were included in the originally submitted supporting information file as Table S1 (Table S4 of the revised supporting information file). The results showed that all the depsipeptides have similar stability to **CP1** in both mouse serum and plasma.

The sentences in the manuscript describing the plasma stability test, “*We also measured mouse plasma stability of other cyclic depsipeptides (DP1–5), and all the cyclic depsipeptides showed high stability (Table S1).*”, was inaccurate and probably a little confusing to the reviewer. Therefore, we have changed the sentences to “*We also measured mouse plasma stability of CP1, DP1, MP1, and other cyclic depsipeptides and N-methylated peptides (DP2–5 and MP2–5). All the cyclic*

depsipeptides and N-methylated peptides showed high stability in mouse plasma (Table S4).”

We have copied Table S4 below for the reviewer’s convenience.

Table S4. Water solubility and mouse plasma stability

compounds	water solubility (μM)	mouse plasma stability (%)
CP1	110 \pm 9	82 \pm 10
DP1	25 \pm 1	81 \pm 8
DP2	56 \pm 4	85 \pm 6
DP3	56 \pm 6	84 \pm 12
DP4	5.5 \pm 0.6	89 \pm 22
DP5	5.2 \pm 0.5	82 \pm 10
MP1	67 \pm 8	108 \pm 10
MP2	91 \pm 6	106 \pm 7
MP3	82 \pm 9	113 \pm 17
MP4	42 \pm 4	105 \pm 2
MP5	4.0 \pm 0.3	120 \pm 13
Cyclosporin A	6 \pm 1	-
DP1L-NH ₂	-	21 \pm 3

Comment from the reviewer

3. Additionally, since one of the primary goals is to derive oral availability for cyclic peptides, it would be interesting to see the stability of these peptides in gastric and intestinal fluids.

>Our response to the comment

We appreciate the valuable comment on the stability assay.

According to the comment, we have evaluated the stability of **CP1**, **DP1** and **MP1** in simulated gastric fluids and intestinal fluids. All the three peptides were stable in both the solutions. Therefore, cyclic depsipeptides have been suggested to be as stable as corresponding cyclic peptides and *N*-methylated peptides in gastric and intestinal fluids.

The following sentences describing the newly conducted stability assay have been added to the manuscript.

[Revised Manuscript p.31]

Stability of **CP1**, **DP1**, and **MP1** in simulated gastric fluid (SGF) and simulated intestinal fluid (SIF) were also examined (Figure 6c and d). Each peptide was incubated in SGF containing pepsin or SIF containing pancreatin for 1–4 h. As a result, no significant degradation of **CP1**, **DP1**, and **MP1** was observed while more than 90% of the control peptides (somatostatin for SGF and oxytocin for SIF) were degraded under the same conditions. These results suggest that **CP1** will be stable against enzymatic degradation until they reach the intestine to be absorbed when they are orally administered, and amide-to-ester substitution and amide *N*-methylation do not decrease the enzymatic stability. Considering that the SGF and SIF assays were conducted at pH 1.2 and pH 6.8, respectively, **CP1**, **DP1**, and **MP1** are also suggested to be stable against hydrolysis at typical pH conditions *in vivo*.

Figure 6. (c) Stability in simulated gut fluid. 98% of a control peptide (somatostatin) was degraded at 4 h under the same conditions. (d) Stability in simulated intestinal fluid. 94% of a control peptide (oxytocin) was degraded at 4 h under the same conditions. Degradation profiles of the control peptides are shown in Figure S49. In (b)–(d), each point represents mean value and standard deviation from experiments carried out in triplicate.

Figure S49. Degradation profiles of control peptides in SGF and SIF

(a) Stability of somatostatin in simulated gut fluid (SGF). (b) Stability of oxytocin in simulated intestinal fluid (SIF).

Comment from the reviewer

- The authors discuss the enhanced permeability of DP2 at higher concentrations due to the involvement of efflux transporters. This should be evaluated by using efflux pump inhibitors.

>Our response to the comment

To confirm the involvement of efflux transporter, we determined efflux ratio, P_e (basolateral-to-apical)/ P_e (apical-to-basolateral), of **DP2** at 1 μM concentration. The efflux ratio was determined to be 27.2, which confirmed the involvement of efflux transporters. The efflux ratio was reduced to 1.05 in the presence of a cocktail of inhibitors against major efflux transporters, further confirming the involvement of efflux transporters.

We have added the following sentences describing the result in the main manuscript.

[Revised manuscript p.10]

The efflux ratio, P_e (basolateral-to-apical)/ P_e (apical-to-basolateral), of **DP2** at 1 μM concentration was determined to be 27.2, confirming the involvement of efflux transporters (Table S2). The permeability value of **DP2** was also measured in the presence of efflux transporter inhibitors (quinidine, sulfasalazine and benzbromarone).³⁵ The efflux ratio was reduced to 1.05 in the presence of the inhibitors (Table S2), further confirming the involvement of efflux transporters and

suggesting the involvement of one or more of the three major efflux transporters in the intestinal epithelium; P-glycoprotein (MDR1/P-gp; ABCB1), breast cancer resistance protein (BCRP; ABCG2) and multidrug-resistance-associated protein 2 (MRP2; ABCC2).

Table S2. Evaluation of the effect of efflux transporters on permeability of DP2 on Caco-2 assay.

	P_e (apical-to-basolateral) ($\times 10^{-6}$ cm/s)	P_e (basolateral-to-apical) ($\times 10^{-6}$ cm/s)	Efflux ratio
Inhibitor (-)	1.5 ± 0.39	40.8 ± 11.0	27.2
Inhibitor (+)*	14.6 ± 3.5	15.3 ± 1.1	1.05

*Caco-2 cells were preincubated with 50 μ M Quinidine, 20 μ M sulfasalazine, and 30 μ M benzbromarone before the assay.

Comment from the reviewer

- It would be worth including a positive control to obtain a feel of the absolute permeability enhancement in the permeability plots.

>Our response to the comment

In the PAMPA result shown in Figure 2c, we have added the permeability of cyclosporin A (0.4×10^{-6} cm/s) as a positive control which is a known membrane permeable peptide.

[Revised manuscript, Figure 2c]

Figure 2. (c) PAMPA and (d) Caco-2 assay of synthesized cyclic peptides. PAMPA was conducted with 2 μ M compounds in PBS containing 5% DMSO and 16 h incubation at 25 °C. Cyclosporin A (CSA) was included as a control for PAMPA (0.4×10^{-6} cm/s). Each bar represents mean value and standard deviation from experiments carried out in quadruplicate. Caco-2 assay was conducted with 1 μ M compounds in HBSS (pH 7.4) containing 10 mM HEPES and 1% DMSO and 3 h incubation at 37 °C. Each bar represents mean value and standard deviation from experiments carried out in triplicate or quadruplicate. The statistical significance of **DP1-5** against **CP1** is shown above the bar of **DP1-5** and the statistical significance of **DP1-5** against **MP1-5** is shown above the bars of **DP1-5** and **MP1-5**. ** $p < 0.01$, * $p < 0.05$. n.s. denotes “not significant”.

Comment from the reviewer

6. I wonder why the PAMPA assay was done for 18 h, as opposed to 4 h. Does it relate to the kinetics of permeation?

>Our response to the comment

As the reviewer pointed out, PAMPA was conducted for 18 h (Figure 1c) or 16 h (Figure 2c) because some peptides exhibit low permeability, and those peptides are difficult to be detected from acceptor wells with 4 h incubation.

Comment from the reviewer

7. Also, can the authors explain the rationale behind the use of various PAMPA assay plates for different libraries?

>Our response to the comment

Thank you for carefully reviewing our manuscript. After receiving this comment from the reviewer, we considered that it would be preferable to do all the PAMPA described in the manuscript on the same type of plate. Therefore, we have re-performed PAMPA in Figure 2c using MultiScreen-IP Filter Plate loaded with 1% lecithin in dodecane. Although the absolute values of permeability of the compounds changed, the trend of the permeabilities among the tested compounds were similar.

Based on the result, we modified the manuscript as follows, but the modification does not affect the major conclusions of this study.

The revised Figure 2c is shown at p.9 of this document.

[Revised manuscript, p.8]

The membrane permeability of the cyclic peptides was initially evaluated using PAMPA (Figure 2c). All the five peptides with amide-to-ester substitutions exhibited significantly higher membrane permeabilities ($P_e = 2.9 \times 10^{-6}$, 2.0×10^{-6} , 1.3×10^{-6} , 4.2×10^{-6} , and 4.4×10^{-6} cm/s for **DP1–5**, respectively) than **CP1** ($P_e = 0.7 \times 10^{-6}$ cm/s) (Figure 2c). Notably, four of the five depsipeptides (**DP1**, **DP2**, **DP3**, and **DP5**) have higher permeabilities than their corresponding *N*-methylated analogues. The permeability enhancement of **DP1** and **DP5** indicates that the membrane permeability of a peptide can be improved by introducing an amide-to-ester substitution on an exposed amide bond. Unexpectedly, amide-to-ester substitutions at unexposed amides (Leu-2, D-Leu-3, and Leu-4) also improved permeability, probably because the substitutions led to the loss of hydrogen bonding networks, which in turn changed the conformational preferences of the cyclic peptides in lipophilic media.

Reviewer 2

Original comments from Reviewer 2

The authors describe a systematic study of the effect of amide-to-ester substitution on the permeability of a series of dipeptides and cyclic hexapeptides. They compare the results with those of the N-methylated analogues. The findings are overall very interesting and highly relevant to the field. The manuscript is also well written. I have some comments for consideration before publication (in the order of appearance in the manuscript).

- While some of the findings should be applicable to other sizes and scaffolds of cyclic peptides, others might be less transferable. Therefore, I would find it important that already the abstract mentions the size of the cyclic peptides in the study (i.e. hexapeptides).

- Introduction: It is mentioned that there is a prevalence of depsipeptides with high permeability (line 80). I think it should also be noted that there are naturally occurring depsipeptides with low permeabilities (e.g. PF1022A, see <https://doi.org/10.1039/D0OB01447H>) and what possible reasons could be for the differences.

- Introduction, line 83: An example for a computational study on the amide-to-ester substitution in protein folding is <https://doi.org/10.1016/j.bbagen.2014.09.014>- Lines 136-139: The NMR experiments of DP2 were again performed in chloroform? Was there any attempt to resolve the multiple conformers of DP2? Given that this peptide might adopt a different permeable conformation than CP1/DP1, it would be useful to determine it.

- Line 146/147: I think the second part of the sentence “DP2 did not show a significant improvement in permeability compared to CP1, as observed in PAMPA (Figure 2d)” is not correct. DP2 shows a higher permeability than CP1 in Figure 2c (as discussed on line 123-125).

- NOEs-MM calculations: Where the NOE restraints applied in a (time/ensemble)-averaged manner? It is not clear to me from the Methods section. The use of instantaneous restraints (i.e. each conformer is expected to fulfil the restraints) is problematic because the NOE-derived distances represent averaged values.

- MD simulations: It would be helpful if the length of the individual trajectories is also mentioned in the main text (or at least the caption of Figure 4) as this is an important information

- What were the input features/dimensions for the PCA analysis? This information is missing, also from the Methods section.

- Figure 4c does not really allow a quantitative comparison of the populations of the different regions. It looks like there is enough simulation data available to construct Markov state models such that the steady-state populations can be estimated. I think this is important for the interpretation of the relative conformational behaviour.

- The authors highlight in the discussion on page 13 (and also in the abstract and discussion on page 17) that “DP1 showed more similar behaviour in conformational transition to CP1 across membrane than MP1.” While this is clearly the case, I do not entirely agree with the discussion on how this is connected to the permeability. CP1 is actually the least permeable of the three, so why should a similar conformational behaviour be beneficial for permeability? That CP1 and DP1 have the same conformational behaviour but different permeability suggests to me that the polarity/lipophilicity is the only relevant factor in the case of these cyclic hexapeptides, reducing the free-energy barrier between the interface and the membrane interior. Looking at Figure 4d and 4c, the same holds true also for MP1, because region A with the permeable conformation is actually significantly populated by all three peptides. The author may want to refine the discussion on page 13 taking these considerations into account.

- Other cyclic peptides, page 15: The results presented here are a bit puzzling to me. The authors write that “all five peptides have only one exposed amide bond (Tyr-6) in the most stable conformation in cyclohexane solution”. However, despite this two of the peptides (CP5 and CP6) have high permeabilities while CP2-CP4 do not. Has this been investigated in the previous studies (Ref. 42,43)? Is there a conformational explanation for these differences? If yes, it would be helpful to summarise them here because this is likely important to understand better the varying effect of the amide-to-ester substitution.

- Proteolytic stability, page 16: Could the authors comment on the (expected) ring-size dependence of the proteolytic stability? While the conformation of the cyclic hexapeptides are compact enough that one can imagine that it is difficult for an enzyme to get access to the ester bond, this “shielding” will likely decrease with increasing ring size. I think this is an important aspect that should be discussed in this section.

- Overall: I would caution a bit regarding the generality of the applicability of the amide-to-ester substitution. The example of DP2 shows clearly that unexpected conformational changes can occur, impacting the permeability either positively or negatively. In addition,

one can expect that the observed proteolytic stability depends on the ring size of the cyclic peptides.

- Methods section, page 21: The details for the steered MD and the PCA analysis should be included. I think the sentence “The membrane permeability of the peptide is estimated based on the slightly modified inhomogeneous solubility-diffusion model (ISDM)” can be removed, because I haven’t seen any such results in the main text.

Point-by-point response to the comments of Reviewer 2

We thank the reviewer for the constructive suggestions. We provide answers to all the reviewer’s comments as follows.

Comment from the reviewer

- While some of the findings should be applicable to other sizes and scaffolds of cyclic peptides, others might be less transferable. Therefore, I would find it important that already the abstract mentions the size of the cyclic peptides in the study (i.e. hexapeptides).

>Our response to the comment

We thank the reviewer for the critical comment. Indeed, the main study was conducted on cyclic hexapeptides only, and the scope of the study should have been stated in the abstract.

After receiving this comment from the reviewer and a related comment from reviewer 3, we attempted to expand the scope of the study. We evaluated the effect of amide-to-ester substitution on permeability of a larger peptide. An amide-to-ester substitution was introduced on octapeptides and a nonapeptide. The introduction of an amide-to-ester substitution on the peptides with larger ring sizes successfully improved membrane permeability which was evaluated with PAMPA. The result indicates that amide-to-ester substitution is a potentially useful strategy to improve membrane permeability of cyclic peptides, not only hexapeptides but also cyclic peptides with larger ring sizes although a larger dataset about large cyclic peptides is desirable in future for better understanding of the scope and limitation of the strategy.

We have modified the abstract to describe the scope of this study more clearly and added the new results about other cyclic peptides in the revised manuscript as described below.

[Revised Abstract]

Naturally occurring peptides with high membrane permeability often have ester bonds on their backbones. However, the impact of amide-to-ester substitutions on the membrane permeability of peptides has not been directly evaluated. Here we, for the first time, report the effect of amide-to-ester substitutions on the membrane permeability and conformational ensemble of cyclic peptides related to membrane permeation. Amide-to-ester substitutions were shown to improve the membrane permeability of dipeptides and a model cyclic hexapeptides. NMR-based conformational analysis and enhanced sampling molecular dynamics simulations suggest that the conformational transition of the cyclic hexapeptide upon membrane permeation is differently influenced by an amide-to-ester substitution and an amide *N*-methylation. The effect of amide-to-ester substitution on membrane permeability of other cyclic hexapeptides, cyclic octapeptides, and a cyclic nonapeptide was also investigated to examine the scope of the substitution. Appropriate utilization of amide-to-ester substitution based on our results will facilitate the development of membrane-permeable peptides.

[Revised manuscript p.28]

Next, we examined the effect of an amide-to-ester substitution on the membrane permeabilities of cyclic peptides with different ring sizes (Figure 5c). **D8.31**, **D8.21**, and **D9.16** are cyclic 8- and 9-mer peptides with multiple *N*-methylated amides that are reported to have high membrane permeabilities. We synthesized the peptides and their derivatives in which an *N*-methylated amide was substituted with a non-*N*-methylated amide (**D8.31-amide**, **D8.21-amide**, and **D9.16-amide**) or ester (**D8.31-ester**, **D8.21-ester**, and **D9.16-ester**) and examined their permeabilities by PAMPA (Figure 5d). In the reported conformations of **D8.31**, **D8.21**, and **D9.16** in CDCl₃, all the *N*-methyl groups are exposed to solvent; therefore, the removal of an *N*-methyl group from the peptides is expected to increase the number of solvent-exposed amide NHs, and **D8.31-amide**, **D8.21-amide**, and **D9.16-amide** show lower

permeability while **D8.31-ester**, **D8.21-ester**, and **D9.16-ester** show higher permeability than **D8.31-amide**, **D8.21-amide**, and **D9.16-amide**. As expected, the original *N*-methylated series (**D8.31**, **D8.21**, and **D9.16**) and the depsipeptides (**D8.31-ester**, **D8.21-ester**, and **D9.16-ester**) showed higher permeability than the non-*N*-methylated peptides (**D8.31-amide**, **D8.21-amide**, and **D9.16-amide**). When the original *N*-methylated peptides and the derivatized depsipeptides are compared, the difference in permeability is sequence-dependent. For the two octapeptides (**D8.31** and **D8.21**), the ester versions showed lower permeabilities than the original *N*-methylated peptides while, for the nonapeptide (**D9.16**), the ester version showed higher permeability than the original *N*-methylated peptide. This is probably because an amide *N*-methylation and an amide-to-ester substitution differently affect the conformational transitions of the cyclic peptides during membrane permeation as demonstrated for **CPI**. These results showed that amide-to-ester substitution is a useful choice for increasing membrane permeability of not only cyclic hexapeptides, but also larger cyclic peptides, although further studies on a more expanded set of large cyclic peptides are desirable in the future to understand the scope and limitation of the substitution.

Figure 5. (c) The structures of **D8.31**, **D8.21**, and **D9.16**, and their derivatives with substitution of an *N*-methylated amide with an amide (**D8.31-amide**, **D8.21-amide**, and **D9.16-amide**) or an ester (**D8.31-ester**, **D8.21-ester**, and **D9.16-ester**). (d) PAMPA of cyclic 8-mer and 9-mer peptides. The enlarged views of the results of **D8.31** series and **D9.16** series are shown in the insets. PAMPA was conducted with 3 μ M compounds in PBS containing 5% DMSO and 16 h incubation at 25 $^{\circ}$ C. Each bar represents mean value and standard deviation from experiments carried out in quadruplicate. ** p < 0.01, * p < 0.05.

Comment from the reviewer

- Introduction: It is mentioned that there is a prevalence of depsipeptides with high permeability (line 80). I think it should also be noted that there are naturally occurring depsipeptides with low permeabilities (e.g. PF1022A, see <https://doi.org/10.1039/D0OB01447H>) and what possible reasons could be for the differences.

>Our response to the comment

In the paper cited by the reviewer, the cyclic depsipeptide PF1022A is shown to be trapped in membrane. This is probably because PF1022A is highly lipophilic and cannot come out from the membrane once it associates with the membrane. This is also an indication of high membrane associating capability of depsipeptides. According to the reviewer's comment, we have cited the paper introduced by the reviewer and added sentences describing that there are also depsipeptides that can be retained in membrane.

[Revised manuscript p.4]

There are also depsipeptides that can be retained in membrane,¹⁷ which also suggests the high membrane associating capability of depsipeptides.

Comment from the reviewer

- Introduction, line 83: An example for a computational study on the amide-to-ester substitution in protein folding is <https://doi.org/10.1016/j.bbagen.2014.09.014>-

>Our response to the comment

We appreciate the recommendation of the reference. We have cited the paper as the reference number 26.

Comment from the reviewer

- Lines 136-139: The NMR experiments of DP2 were again performed in chloroform?

>Our response to the comment

The solvent information of the NMR experiments of **DP2** was missing as the reviewer pointed out. We have added a sentence in the manuscript to include the solvent information.

[Revised manuscript p.9]

From the NMR spectra in CDCl₃, stable conformations of **DP2** in lipophilic environment were determined.

Comment from the reviewer

- Was there any attempt to resolve the multiple conformers of DP2? Given that this peptide might adopt a different permeable conformation than CP1/DP1, it would be useful to determine it.

>Our response to the comment

After receiving the comments from the reviewer and a related comment from reviewer 3, we have resynthesized **DP2** and measured NMR spectra of **DP2** again. The NMR spectra have turned out to have a single set of peaks. The multiple peaks, observed in the original manuscript, might have appeared due to modification of **DP2** or contamination of impurities.

Using the newly obtained NMR spectra, we have investigated the stable conformation of **DP2** in CDCl₃. The **DP2** has been shown to have a different intramolecular hydrogen bonding network from that of **CP1**. One of the solvent-exposed amide hydrogens in **CP1** was found to form a new hydrogen bond in **DP2**. This may explain why **DP2** shows higher membrane permeability than **CP1**.

We have added sentences describing the stable conformation of **DP2** and the comparison of the conformation with the stable conformation of **CP1** as follows.

[Revised manuscript p.9]

From the NMR spectra in CDCl₃, stable conformations of **DP2** in the lipophilic

environment were determined. Interestingly, the amide NH of Tyr-1 residue in the most stable conformations of **DP2** forms an intramolecular hydrogen bond with the carbonyl oxygen of the Leu-5 residue (Figure S7a) while the same amide NH was reported to be solvent-exposed in **CP1**³² (Figure S7b). The amide-to-ester substitution on the amide that forms an intramolecular hydrogen bond in **CP1** caused the rearrangement of the intramolecular hydrogen bonding network, which is assumed to be the reason for the unexpectedly improved membrane permeability of **DP2**. The unexpected improvement of membrane permeability of **DP3** and **DP4** is also probably due to the conformational changes upon the amide-to-ester substitution as seen in **DP2**.

Figure S7. Comparison of the most stable conformations of DP2 and CP1.

The most stable conformations of (a) **DP2** and (b) **CP1** from NMR analysis are shown. Intramolecular hydrogen bonds are highlighted by dotted blue lines. Shown in (b) is the most stable conformation from our NMR analysis (Figure 3a and Figure S9–14), which is similar to the reported conformation in the previous report (Rezai, T., *et al.*, *J. Am. Chem. Soc.* **128**, 2510–2511 (2006)).

Comment from the reviewer

- Line 146/147: I think the second part of the sentence “DP2 did not show a significant improvement in permeability compared to CP1, as observed in PAMPA (Figure 2d)” is not correct. DP2 shows a higher permeability than CP1 in Figure 2c (as discussed on line 123-125).

>Our response to the comment

We appreciate the careful reading of the manuscript. The sentence was incorrect as the reviewer pointed out. We have revised the sentence as follows.

[Revised manuscript, p10]

Unlike the observation in PAMPA, **DP2** did not show a significant difference in permeability compared to **CP1** (Figure 2d).

Comment from the reviewer

- NOEs-MM calculations: Where the NOE restraints applied in a (time/ensemble)-averaged manner? It is not clear to me from the Methods section. The use of instantaneous restraints (i.e. each conformer is expected to fulfil the restraints) is problematic because the NOE-derived distances represent averaged values.

>Our response to the comment

In the originally submitted manuscript, the stable conformations were extracted from the conformations generated by a molecular mechanics-based conformational search. Penalties were given to the generated conformations based on the consistency with the experimental data. NOE signals were converted to proton-proton distances based on the signal intensities. The conformers with proton-proton distances that differ more than ± 1.5 Å from the NOE-derived distances are given penalties. Similarly, the conformers with dihedral angles that differ more than $\pm 15^\circ$ from the 3J -derived dihedral angles are given penalties. Based on the relative energy from the molecular mechanics calculations and penalties derived from the experiments, the conformations were ranked, and the stable conformations were determined. Considering that the NOE-derived distances were applied as restraints with ± 1.5 Å width, we believe that the determined stable conformations are reliable.

To confirm that the stable conformations of the peptides generated by the NOEs-MM are reliable, we have also experimentally determined the peptide structures solely by NMR constraints using Xplor-NIH with commonly used simulated annealing protocols for structure determination of proteins and peptides. In this analysis, NOE signals are treated in a semi-quantitative manner giving the certain allowance NOT to be too precise, since observed NOE signals are time/ensemble-

averaged. In the calculation, NOE signals are divided into three categories: strong, medium, and weak. Each of the categories is given a range of proton-proton distance, i.e., 1.8–2.7 Å, 1.8–3.5 Å, and 1.8–5.0 Å, respectively. 3J -derived dihedral angles are converted to dihedral angles using a Karplus equation and the values were given $\pm 30^\circ$ allowance. The generated conformations of **CP1**, **MP1**, and **DP1** were consistent with the conformations on the originally submitted manuscript. RMSD values between the originally submitted structure and revised structure were 0.250, 0.529, 0.489 Å for **CP1**, **MP1**, and **DP1**, respectively

The stable conformations of **CP1** and **DP1** in Figure 3 were replaced with the newly determined conformations. With the replacement of the conformations, the RMSD value between **CP1** and **DP1** and that between **CP1** and **MP1** were changed. The changes made for the conformational analysis do not influence the major conclusions drawn from the experiments in the manuscript. In addition, it is difficult and rather imprecise to uniquely determine the intramolecular hydrogen bonding network of each peptide from the conformational ensemble. Therefore, the dashed lines indicating the hydrogen bonding networks in Figure 3 have been removed.

[Revised manuscript, p14]

The root-mean-square deviation (RMSD) of their backbones was calculated to be 0.426, which confirmed that the amide-to-ester substitution of the exposed amide NH does not significantly change the solution conformations of the cyclic hexapeptide in a membrane-like lipophilic environment.

[Revised manuscript, p15]

The RMSD between the backbones of **MP1** and **CP1** was 0.782. The value is small but higher than the RMSD between **DP1** and **CP1**. These results indicate that **MP1** forms a conformational state that is similar but a little different from that of **CP1** in a lipophilic environment (Figures S21–27).

Figure 3. Stereoviews of NMR solution structures of (a) **CP1** and (b) **DP1** in CDCl_3 . (c) The superposition of **DP1** with **CP1**. **DP1** is shown in brown and **CP1** is shown in blue.

Comment from the reviewer

- MD simulations: It would be helpful if the length of the individual trajectories is also mentioned in the main text (or at least the caption of Figure 4) as this is an important information

>Our response to the comment

We have revised the main text as follows to describe the length of the simulations.

[Revised manuscript, p18]

An MD simulation was performed for 300 ns after a 200 ns equilibration process for each peptide.

Comment from the reviewer

- What were the input features/dimensions for the PCA analysis? This information is missing, also from the Methods section.

>Our response to the comment

We have revised our description of the principal components analysis procedure in the main text as follows.

[Revised manuscript, p18]

In this analysis, 72-dimensional eigenvectors were calculated from the variance-covariance matrices constructed from 3D coordinates of the backbone atoms of residues 2–6, which are common to **CP1**, **DP1**, and **MP1**.

We have also added the description of the principal component analysis in the method section as follows.

[Revised manuscript, p41]

Peptide conformations were analyzed by principal component analysis based on the 3D coordinates of the backbone atoms of residues 2–6 (24 atoms), the common chemical structure of **CP1**, **DP1**, and **MP1**. 72-dimensional eigenvectors were obtained by diagonalizing the variance-covariance matrix calculated from all snapshots of the three peptides (630,000 snapshots) with the temperature of 300 K superimposed on the 2–6 residues of the backbone atoms. Subsequently, trajectories corresponding to the inside, at the interface, and outside the lipid membrane of each peptide were projected onto the eigenvectors corresponding to the first and second principal components.

Comment from the reviewer

- Figure 4c does not really allow a quantitative comparison of the populations of the different regions. It looks like there is enough simulation data available to construct Markov state models such that the steady-state populations can be estimated. I think this is important for the interpretation of the relative conformational behaviour.

>Our response to the comment

Because our calculations are based on umbrella sampling, the data in Figure 4, e.g., 0–6 Å, are a mixture of data constrained to slightly different positions. Those molecules feel slightly different forces from the membrane and solvent molecules. In addition, the center of gravity of the nitrogen atoms in the backbone is always harmonically restrained at certain position. It is not clear how the effects of these constraints affect the estimation of transition probabilities between states. On the other hand, the distribution of the obtained conformations is expected to be reasonably

accurate, since the distribution is estimated based on an effective sampling protocol. Therefore, it is expected that the population of each conformation can be compared quantitatively well by integrating the histograms projected into PCA space for each specific region. We have added the information about the percentage of each conformation in Figure S47. In addition, we have added the procedure to calculate the percentages for each conformation in the Methods section as follows.

[Revised manuscript, p42]

After that, the representative conformations, labeled as A to E, were extracted from the peaks of the distributions. Then, the percentage of all conformations in a rectangular region in two-dimensional PCA space around the representative conformation is estimated (Figure S47). The rectangular region in PCA space containing conformations A–D was defined as the region containing conformations within 0.4 Å of the RMSD of the backbone atoms of residues 2–6 from the representative structure in the trajectory of the interface of **DP1**. In calculating the percentage of conformations A–D, the rectangular region on the PCA space defined here was adopted for all peptides. The rectangular region in PCA space containing conformations E was defined as the region containing conformations within 0.4 Å of the RMSD of the backbone atoms of residues 2–6 from the representative structure in the trajectory of the outside of the membrane of **MP1**.

Figure S47. The percentage of all conformations in each region in the two-dimensional PCA space from the MD simulations

Conformational ensembles of **CP1**, **DP1**, and **MP1** inside, at the interface, and outside the lipid

membrane projected onto the first and second principal axes. The percentages of the major conformations are described.

Comment from the reviewer

- The authors highlight in the discussion on page 13 (and also in the abstract and discussion on page 17) that “DP1 showed more similar behaviour in conformational transition to CP1 across membrane than MP1.” While this is clearly the case, I do not entirely agree with the discussion on how this is connected to the permeability. CP1 is actually the least permeable of the three, so why should a similar conformational behaviour be beneficial for permeability? That CP1 and DP1 have the same conformational behaviour but different permeability suggests to me that the polarity/lipophilicity is the only relevant factor in the case of these cyclic hexapeptides, reducing the free-energy barrier between the interface and the membrane interior. Looking at Figure 4d and 4c, the same holds true also for MP1, because region A with the permeable conformation is actually significantly populated by all three peptides. The author may want to refine the discussion on page 13 taking these considerations into account.

>Our response to the comment

We believe that, in the manuscript, there are no sentences where the similarity of the conformational transitions between **CP1** and **DP1** is directly connected to the permeability. Rather, we intended to attribute the reason for the increased membrane permeability of **DP1** to the increased lipophilicity of the conformations of **DP1** in region A compared with the conformations of **CP1** in region A, which is the same with the reviewer’s opinion.

If the reviewer considers there are any logically inappropriate discussions in particular sentences, we’d be happy to reconsider the discussion.

Comment from the reviewer

- Other cyclic peptides, page 15: The results presented here are a bit puzzling to me. The authors write that “all five peptides have only one exposed amide bond (Tyr-6) in the

most stable conformation in cyclohexane solution”. However, despite this two of the peptides (CP5 and CP6) have high permeabilities while CP2-CP4 do not. Has this been investigated in the previous studies (Ref. 42,43)? Is there a conformational explanations for these differences? If yes, it would be helpful to summarise them here because this is likely important to understand better the varying effect of the amide-to-ester substitution.

>Our response to the comment

According to the previous study (Ono, S. *et al.* Conformation and Permeability: Cyclic Hexapeptide Diastereomers. *J. Chem. Inf. Model.* **59**, 2952–2963 (2019)), the membrane permeability of cyclic peptides has a high correlation with the solvent accessible surface area (SASA) of peptides. Indeed, although the number of solvent-exposed amide hydrogen is the same among CP2–CP6, CP2–CP4 that have low-to-medium permeability have higher SASA than CP5 and CP6 that have high permeability.

The introduction of an amide-to-ester substitution on an exposed amide is expected to similarly affect the SASA on all the peptides, yet the effect of the substitution was different among CP2–CP6. As described in the manuscript, we consider that the excess lipophilicity introduced on CP5 and CP6 is one reason for the decreased membrane permeability. According to another previous report, amide NH of Tyr-1 residue of CP5 and CP6 may be involved in hydrogen bondings. Therefore, conformational changes caused by the substitution might be another reason. A description about this possible reason has been added to the manuscript as follows.

[Revised manuscript, p.27]

Another possible reason is that the amide NH of the Tyr-1 residue of CP5 and CP6 is involved in intramolecular hydrogen bondings as suggested from previous conformational studies in CDCl₃⁴⁸ and the substitution caused conformational changes on the cyclic peptides.

Comment from the reviewer

- Proteolytic stability, page 16: Could the authors comment on the (expected) ring-size dependence of the proteolytic stability? While the conformation of the cyclic

hexapeptides are compact enough that one can imagine that it is difficult for an enzyme to get access to the ester bond, this “shielding” will likely decrease with increasing ring size. I think this is an important aspect that should be discussed in this section.

>Our response to the comment

We agree that the compact ring size of the cyclic hexapeptide contributes to the shielding effect on the ester bond. Therefore, the high enzymatic stability of the cyclic depsipeptides as well as cyclic peptides may be ring-size dependent. We have added a sentence to describe this point on the manuscript.

[Revised manuscript, p31]

The effect of macrocyclization on shielding an ester bond from enzymatic degradation may depend on the sequence and the ring size of cyclic peptides. Such sequence/ring size dependence on the enzymatic stability is an interesting subject of a future study of cyclic depsipeptides.

Comment from the reviewer

- Overall: I would caution a bit regarding the generality of the applicability of the amide-to-ester substitution. The example of DP2 shows clearly that unexpected conformational changes can occur, impacting the permeability either positively or negatively. In addition, one can expect that the observed proteolytic stability depends on the ring size of the cyclic peptides.

>Our response to the comment

We appreciate the critical comment.

We do not intend to claim that amide-to-ester substitution always preserves the conformational states of cyclic peptides and increases membrane permeability of the peptides. Rather, in this study, we would like to show that amide-to-ester substitution is potentially useful for increasing membrane permeability of cyclic peptides especially when the substitution is introduced on an amide whose hydrogen is exposed to solvent.

However, there were sentences where the utility of an amide-to-ester substitution is over-generalized, as the reviewer pointed out. Therefore, we have modified the manuscript. First, the abstract has been changed to clarify that the study is mostly conducted on a model cyclic hexapeptide. (The revised abstract is shown in p.15 of this document.) Second, we have removed the sentence “(2) a cyclic peptide with an amide-to-ester substitution of an exposed amide bond can adopt conformations in lipophilic media that are similar to those of the original peptide” from the conclusion section because it is not the general feature of amide-to-ester substitution. Third, we have changed the sentence “an amide-to-ester substitution had a smaller influence on conformational transitions during the membrane permeation process of a cyclic peptide than an amide N-methylation for the cyclic hexapeptide *CPI*” to “an amide-to-ester substitution and an amide N-methylation differently influence on the conformational transitions during the membrane permeation process of a cyclic peptide and the difference in conformational transitions influences the membrane permeability” in the conclusion section.

We believe that the applicability of amide-to-ester substitution for increasing membrane permeability are not too much generalized in the revised version of the manuscript. However, if the reviewer considers any sentences in the manuscript are still not suitable or any additional sentences need to be added, we'd be happy to consider further revising the manuscript.

Comment from the reviewer

- Methods section, page 21: The details for the steered MD and the PCA analysis should be included. I think the sentence “The membrane permeability of the peptide is estimated based on the slightly modified inhomogeneous solubility-diffusion model (ISDM)” can be removed, because I haven't seen any such results in the main text.

>Our response to the comment

We have added the description of the steered MD in the method section as follows.

[Revised manuscript, p41]

The initial coordinates are extracted from the trajectory of pre-executed steered MD⁶⁴ with solute tempering.⁶⁵ The center of mass of the nitrogen atoms of peptide bonds was pulled from $z = 40.0 \text{ \AA}$, a position slightly beyond the reaction coordinate for the REST/REUS simulation, to -5.0 \AA , a position slightly beyond the center of the membrane. A pulling rate of 0.25 \AA/ns and force constant of $3.0 \text{ kcal/mol/\AA}^2$ were used. In this process, the temperature of the peptide is set to $2,100 \text{ K}$ to obtain diverse conformations.

We have also added the description of the principal component analysis in the Method section as follows.

[Revised manuscript, p41]

Peptide conformations were analyzed by principal component analysis based on the 3D coordinates of the backbone atoms of residues 2–6 (24 atoms), the common chemical structure of **CPI**, **DP1**, and **MP1**. 72-dimensional eigenvectors were obtained by diagonalizing the variance-covariance matrix calculated from all snapshots of the three peptides (630,000 snapshots) with the temperature of 300 K superimposed on the 2–6 residues of the backbone atoms. Subsequently, trajectories corresponding to the inside, at the interface, and outside the lipid membrane of each peptide were projected onto the eigenvectors corresponding to the first and second principal components.

The sentence “*The membrane permeability of the peptide is estimated based on the slightly modified inhomogeneous solubility-diffusion model (ISDM)*” in the Method section has been deleted.

Reviewer 3

Original comments from Reviewer 3

This paper explores the effect of replacing amide and n-methylated amides from a highly studied and reported cyclic hexapeptide system with esters (depsipeptides) on membrane and cell permeability. The authors show that in some cases this replacement resulted in improved permeability across both artificial membranes and cell monolayers. The authors claim that the depsipeptides do not affect the three-dimensional structures of these peptides as much as n-methylation.

Whilst this work is of some limited interest in the field, the scope of what is presented is restricted to largely one cyclic hexapeptide system and some of these authors (and others) have published on amide to ester substitutions in small molecule systems that improve permeability, hence limiting the novelty of this work. <https://pubs.acs.org/doi/10.1021/acs.jmedchem.1c01496>

I don't consider that paper would appeal to a broad scientific audience and is better suited to a more specialized journal. However, to do that the authors still need to do a lot of work to correct some major flaws and provide missing support for suggested findings.

Specific Comments

Section 1- Dipeptides

The Authors state that for all sequences the depsipeptides exhibited the highest permeability, but Figure 1c shows that this is only clearly the case for D1. No statistical analysis of D2 vs M2 or D3 vs M3 is provided, and there does not appear to be much difference in these numbers. Importantly, these compounds have only very low permeability. How confident are the authors of the accuracy of the quantification of such small amounts of these peptides in the acceptor/donor wells?

The scope of this experiment is very limited. Why did the authors only choose to look at dipeptides? Other physical properties of those dipeptides such as LogP, HPLC elution times, LogD and solvation energy could also be analyzed.

Section 2 -Cyclic Peptides

Figure 2c and 2d- Again no statistical analysis is provided. Are all 5 comparisons statistically significant?

Figure 2f- No error bars at all are on this curve CP50 3.4 vs 7.4 μM are reported with no error of measurement or statistical significance. Are these numbers actually different?

Figure 2g- The confocal microscopy appears to show a much larger difference than the flow cytometry results, which are virtually identical. Why is this the case?

The explanation of structural heterogeneity for DP2 (from line 132 to line 142) by NMR spectroscopy is inconclusive and confusing. Multiple sets of NMR signals for Tyr-1 could be due to impurity. If multiple sets of NMR signals are the result of exchangeable structures of DP2 in solution, then variable temperature and/or solution NMR experiments should be carried out to experimentally support this claim.

Another confusing statement from the authors is that there are three sets of NMR resonances for Tyr-1 H_n and H_b, but only two sets of peaks for H_c (lines 137 to 138). The H_n and H_c of Tyr-1 should have the same number of sets of signals.

There are inconsistent results for CP1 cell permeability as figure 2c and 2d indicated CP1 has very low membrane and cell permeability. However figure 2e, 2f and 2g showed CP1 with chloroalkane tag can significantly penetrate through the cell membrane, is this result due to the tag itself entering the cell or enhancing permeability?

Section 3-

There are no NOE summaries, amide coupling constants, variable temperature amide chemical shifts included for solving 3D structures in the SI. These data are absolutely essential for investigating and verifying any solution structure (e.g. CP1, DP1 and MP1), and for reviewing this paper.

Figure S11. ROESY spectrum for CP1 does not show the crucial inter-residue ROEs

Figure S17. ROESY spectrum for DP1 shows overlapping alpha proton signals for residues l3, L4 and p6. There was no discussion of how the authors derived ROEs from those overlapping signals. This ROESY spectrum also does not show any inter-residue ROEs, suggesting that the structure derived from it is unreliable.

Only AlogP was used to calculate predicted 'lipophilicity' - why not any other calculations? Do other calculations lead to the same predictions for amide to ester replacements? AlogP often does not accurately predict experimental logP data.

Section 4-

The in silico simulation is the strongest part of this work. This is why this work may be better suited for a computational chemistry journal.

It is crucial that modeling results in this section are supported by at least some experimental data (e.g. CD, NMR amide proton temperature dependence, amide coupling constants to name just a few). Modeling predictions are notoriously prone to erroneous

interpretations when not supported by strong experimental evidence of structure to prove the model.

Section 5-

I find the title of this section misleading as only cyclic hexapeptides with the sequence [PLLLLY] were used. As such the authors' claims of being able to improve permeability in a range of peptides is misleading and actually pretty limited. No n-methyl comparisons were used in this section.

This paper would be greatly enhanced if ester replacements were made to a broader range of peptides and the effects on permeability and structure explored.

Again no statistical analysis.

Section 6-

Stability of peptides in different pH buffers should also be examined, since depsipeptides are well known to be prone to hydrolysis. Indeed, that is a major reason why depsipeptides have not translated well in medicinal chemistry - in most cases they tend to be quite unstable in vivo and prone to cleavage.

Discussion

Figure 6 - this figure is an oversimplification, the effect will not be just because of one water interaction.

Methods

PAMPA assay: Different assay systems were used for CP1 and derivatives than for the other peptides. Why?

The Genetest Pre-coated PAMPA plate can give very different results than the homemade system. All compounds need to be measured in the same system to make valid comparisons.

NMR measurements: Spectra are recorded in CDCl₃ but DMSO appears in some of the spectra. Does the presence of this more polar solvent affect the structures presented?

Authors: Why so many corresponding authors?

Point-by-point response to the comments of Reviewer 3

We thank the reviewer for the constructive suggestions. We provide answers to all the reviewer's comments as follows.

Comment from the reviewer

This paper explores the effect of replacing amide and n-methylated amides from a highly studied and reported cyclic hexapeptide system with esters (depsipeptides) on membrane and cell permeability. The authors show that in some cases this replacement resulted in improved permeability across both artificial membranes and cell monolayers. The authors claim that the depsipeptides do not affect the three-dimensional structures of these peptides as much as n-methylation.

Whilst this work is of some limited interest in the field, the scope of what is presented is restricted to largely one cyclic hexapeptide system and some of these authors (and others) have published on amide to ester substitutions in small molecule systems that improve permeability, hence limiting the novelty of this work.
<https://pubs.acs.org/doi/10.1021/acs.jmedchem.1c01496>

>Our response to the comment

We appreciate the reviewer's critical comment. As the reviewer pointed out, the originally submitted manuscript described mostly only about one cyclic hexapeptide with hydrophobic side chains. Therefore, the scope of the amide-to-ester substitution for improving membrane permeability of peptides appeared to be limited.

To further investigate the scope of the amide-to-ester substitution for improving membrane permeability, the effect of amide-to-ester substitutions on membrane permeability of cyclic hexapeptides with hydrophilic side chains and cyclic peptides with larger ring sizes (8- and 9-mer peptides) have been evaluated.

We have synthesized and measured the membrane permeability of the new cyclic peptides and their derivatives with an amide-to-ester substitution. The result showed that an amide-to-ester substitution has a potential to improve membrane permeability of cyclic peptides, not only hexapeptides with hydrophobic side chains but also, cyclic peptides with hydrophilic side chains and larger ring sizes.

With these new results, we argue that the amide-to-ester substitution strategy

for improving membrane permeability is not only applicable to a peptide with a specific ring size and sequence but also applicable to other cyclic peptides.

Although there is a preceding paper regarding the effect of amide-to-ester substitution on small molecule system as the reviewer pointed out, our work is the first to directly evaluate the effect of the introduction of an amide-to-ester substitution to the backbone amide bond of peptides. (Please also note that the preprint of this work (<https://doi.org/10.26434/chemrxiv.12272861.v1>) has been posted on ChemRxiv, before the paper cited by the reviewer was published.)

Large molecules like the cyclic peptides investigated in this study form complex and dynamic three-dimensional structures, and the membrane permeability depends on the three-dimensional structures. The complexity and dynamic nature of the structure make the control of membrane permeabilities of cyclic peptides much more difficult than those of small molecules. The present work provides insights into how the substitution affects the conformational dynamics of the cyclic peptides and how the conformational changes influence on the membrane permeability. Therefore, we believe the present work provides a significant scientific advancement from the previous reports.

The comparison of the effect of an amide-to-ester substitution and amide *N*-methylation is another highlight of this study. We have shown that amide-to-ester substitution sometimes increases membrane permeability of peptides more than amide *N*-methylation although it is sequence dependent. Besides, the conformational changes of peptides, depsipeptide and an *N*-methyl peptide, during the membrane permeation process have been investigated for the first time by MD simulations and NMR experiments. These results indicate that, in addition to *N*-methylated peptides, depsipeptides may be a new option for the design of membrane-permeating peptide.

The new results and discussion about the significance and limitation of this study are included in the revised manuscript as follows. Besides, we have modified the abstract to describe the scope of this study more clearly and added the new experimental results about other cyclic peptides in the revised manuscript as shown in the following pages.

[Revised manuscript p.27]

We also examined the amide-to-ester substitution strategy for increasing the membrane permeability of cyclic peptides with a hydrophilic residue. **CP1** derivatives in which Tyr-1 residue is substituted with Phe residue and Leu-2 residue is substituted to Ser or Lys (**CP1-Y1F-L2S** and **CP1-Y1F-L2K**) (Figure 5a). These two peptides were further modified at the Phe-1 residue with an amide-to-ester substitution or an amide *N*-methylation. The membrane permeability of these peptides was examined by PAMPA. The membrane permeability of **CP1-Y1F-L2S** was significantly increased by an amide-to-ester substitution while the permeability was not largely increased by an amide *N*-methylation (Figure 5b left). The permeability value of **CP1-Y1F-L2K** was also increased by an amide-to-ester substitution (Figure 5b right). Although the permeabilities of the original **CP1-Y1F-L2K** and the **CP1-Y1F-L2K** with an *N*-methylamide were under the quantification limits, the differences of the permeabilities with that of the **CP1-Y1F-L2K** with an ester were statistically significant considering the quantification limits (7.0×10^{-10} cm/s for the original **CP1-Y1F-L2K** and 7.5×10^{-10} cm/s for the **CP1-Y1F-L2K** with an *N*-methylamide). However, the permeability value of **CP1-Y1F-L2K** with an ester is still low (4.9×10^{-9} cm/s), and therefore, further modifications, such as *N*-alkylation^{29,48} is desirable for practical applications of peptides with charged residues like **CP1-Y1F-L2K**.

Next, we examined the effect of an amide-to-ester substitution on the membrane permeabilities of cyclic peptides with different ring sizes (Figure 5c). **D8.31**, **D8.21**, and **D9.16** are cyclic 8- and 9-mer peptides with multiple *N*-methylated amides that are reported to have high membrane permeabilities. We synthesized the peptides and their derivatives in which an *N*-methylated amide was substituted with a non-*N*-methylated amide (**D8.31-amide**, **D8.21-amide**, and **D9.16-amide**) or ester (**D8.31-ester**, **D8.21-ester**, and **D9.16-ester**) and examined their permeabilities by PAMPA (Figure 5d). In the reported conformations of **D8.31**, **D8.21**, and **D9.16** in CDCl₃, all the *N*-methyl groups are exposed to solvent; therefore, the removal of an *N*-methyl group from the peptides is expected to increase the number of solvent-exposed amide NHs, and **D8.31-amide**, **D8.21-amide**, and **D9.16-amide** show lower permeability while **D8.31-ester**, **D8.21-ester**, and **D9.16-ester** show higher permeability than **D8.31-amide**, **D8.21-amide**, and **D9.16-amide**. As expected, the

original *N*-methylated series (**D8.31**, **D8.21**, and **D9.16**) and the depsipeptides (**D8.31-ester**, **D8.21-ester**, and **D9.16-ester**) showed higher permeability than the non-*N*-methylated peptides (**D8.31-amide**, **D8.21-amide**, and **D9.16-amide**). When the original *N*-methylated peptides and the derivatized depsipeptides are compared, the difference in permeability is sequence-dependent. For the two octapeptides (**D8.31** and **D8.21**), the ester versions showed lower permeabilities than the original *N*-methylated peptides while, for the nonapeptide (**D9.16**), the ester version showed higher permeability than the original *N*-methylated peptide. This is probably because an amide *N*-methylation and an amide-to-ester substitution differently affect the conformational transitions of the cyclic peptides during membrane permeation as demonstrated for **CP1**. These results showed that amide-to-ester substitution is a useful choice for increasing membrane permeability of not only cyclic hexapeptides, but also larger cyclic peptides, although further studies on a more expanded set of large cyclic peptides are desirable in the future to understand the scope and limitation of the substitution.

Figure 5. The effect of an amide-to-ester substitution on cyclic hexapeptides with a hydrophilic residue and cyclic 8- and 9-mer peptides. (a) The structures of **CP1** derivatives with a hydrophilic residue and their derivatives with an amide-to-ester substitution or an amide *N*-methylation. (b) The membrane permeabilities of the **CP1** derivatives shown in Figure 5a. N.D. denotes “Not Detected”. (c) The structures of **D8.31**, **D8.21**, and **D9.16**, and their derivatives with substitution of an *N*-methylated

amide with an amide (**D8.31-amide**, **D8.21-amide**, and **D9.16-amide**) or an ester (**D8.31-ester**, **D8.21-ester**, and **D9.16-ester**). (d) PAMPA of cyclic 8- and 9-mer peptides. The enlarged views of the results of **D8.31** series and **D9.16** series are shown in the insets. PAMPA was conducted with 3 μM compounds in PBS containing 5% DMSO and 16 h incubation at 25 °C. Each bar represents mean value and standard deviation from experiments carried out in quadruplicate. ** $p < 0.01$, * $p < 0.05$.

[Revised manuscript p.34]

Cyclic peptides form complex three-dimensional structures that dynamically change in a short time scale. Therefore, understanding and controlling the membrane permeability of cyclic peptides present unique challenges.

The present study on the conformations and physicochemical properties of a model cyclic hexapeptide and its derivatives with amide-to-ester substitution indicates the following two intriguing insights on features of amide-to-ester substitutions: (1) By substituting an amide bond exposed in a lipophilic environment to an ester bond, the membrane permeability of the cyclic peptide can be improved, (2) the amide-to-ester substitution of cyclic peptides does not decrease proteolytic stability as long as the substitution is introduced on cyclic hexapeptides. The NMR-based conformational analysis and enhanced sampling MD simulations suggest the dynamic conformational transition of the cyclic peptides among “open” and “closed” conformations upon permeation across lipid bilayer membrane. The conformational transition of **DPI** across the membrane is similar to that of the original peptide **CPI**. Interestingly, an amide-to-ester substitution increased the membrane permeability of the cyclic hexapeptides more than amide *N*-methylation in some of the tested cases.

In the present study, we have also examined the scope of an amide-to-ester substitution for increasing the membrane permeability of larger cyclic peptides. The result suggests that the substitution can increase membrane permeability of cyclic 8- and 9-mer peptides although it is sequence-dependent. Further studies on the effects of an amide-to-ester substitution on membrane permeability as well as conformational ensembles of large cyclic peptides are expected to facilitate more strategic applications of the substitution for increasing membrane permeability of cyclic peptides.

[Revised Abstract]

Naturally occurring peptides with high membrane permeability often have ester bonds on their backbones. However, the impact of amide-to-ester substitutions on the membrane permeability of peptides has not been directly evaluated. Here we, for the first time, report the effect of amide-to-ester substitutions on the membrane permeability and conformational ensemble of cyclic peptides related to membrane permeation. Amide-to-ester substitutions were shown to improve the membrane permeability of dipeptides and a model cyclic hexapeptides. NMR-based conformational analysis and enhanced sampling molecular dynamics simulations suggest that the conformational transition of the cyclic hexapeptide upon membrane permeation is differently influenced by an amide-to-ester substitution and an amide *N*-methylation. The effect of amide-to-ester substitution on membrane permeability of other cyclic hexapeptides, cyclic octapeptides, and a cyclic nonapeptide was also investigated to examine the scope of the substitution. Appropriate utilization of amide-to-ester substitution based on our results will facilitate the development of membrane-permeable peptides.

Comment from the reviewer

- Section 1- Dipeptides

The Authors state that for all sequences the depsipeptides exhibited the highest permeability, but Figure 1c shows that this is only clearly the case for D1. No statistical analysis of D2 vs M2 or D3 vs M3 is provided, and there does not appear to be much difference in these numbers. Importantly, these compounds have only very low permeability. How confident are the authors of the accuracy of the quantification of such small amounts of these peptides in the acceptor/donor wells?

>Our response to the comment

We have added the information of statistical significance in Figure 1c. There are statistically significant differences between **D2** and **M2**, and **D3** and **M3**.

Using LC-MS, cyclic peptides can be detected at low concentrations. For example, dipeptide **P3**, which has the lowest permeability among the tested peptides, can be quantitated at 0.01 μ M at the lowest. 0.02 μ M of **P3** was detected in the assay shown on Figure 1 which is higher than the limit of quantification. The reliability of

the measurement is also indicated by the small error bars.

For the reviewer's reference, we show the revised Figure 1c with the statistical analysis and the standard curve and the chromatograms of the **P3** from the acceptor wells after PAMPA (Reviewer only Figure) below.

[Revised manuscript, Figure 1]

Figure 1. (a) General structures of model dipeptides. (b) Sequences of synthesized dipeptides. (c) Permeability values of the synthesized dipeptides measured by PAMPA. PAMPA was conducted with 10 μ M compounds in 5% DMSO/PBS (pH 7.4) and 18 h incubation at 25 °C. Each bar represents the mean value and the standard deviation from experiments carried out in quadruplicate. ** $p < 0.01$

Reviewer only Figure. The standard curve of **P3** (left) and selected ion chromatograms of **P3** from acceptor wells in PAMPA shown on Figure 1c.

Comment from the reviewer

- The scope of this experiment is very limited. Why did the authors only choose to look at dipeptides? Other physical properties of those dipeptides such as LogP, HPLC elution times, LogD and solvation energy could also be analyzed.

>Our response to the comment

We used dipeptide as a minimal model of peptides to discuss the effect of amide-to-ester substitution on membrane permeability.

According to the reviewer's suggestion, we have calculated CLogP and ALogP values and measured UPLC retention times and $\log D_{\text{dec/w}}$ values (Table S1). All the values were higher for **D1-3** than the corresponding **M1-3**. In addition, CLogP, ALogP and UPLC retention time were also higher for **D1-3** than the corresponding **M1-3** while the $\log D_{\text{dec/w}}$ values were not significantly different between **D1-3** and **M1-3**. These results indicate that the high membrane permeability of depsipeptides is at least partly derived from higher lipophilicity of ester compared with amide and *N*-methanamide.

The reviewer also recommended us to analyze the solvation energies. However, we have considered that reliable estimation of solvation energy is difficult for dipeptides because dipeptides are conformationally flexible. Therefore, we have not determined the solvation energies.

The new additional data (CLogP, ALogP, UPLC retention time, and $\log D_{\text{dec/w}}$) have been added as Table S1, and the results and discussion about the new data have been described in the revised manuscript as follows.

[Revised manuscript p.6]

The higher membrane permeability of **D1-3** compared with **P1-3** can be attributed to the higher lipophilicity of ester compared with amide. Calculated distribution coefficients (CLogP and ALogP), retention time on an octadecyl column during liquid chromatography, and experimental distribution coefficients between decadiene and aqueous buffer ($\log D_{\text{dec/w}}$) were determined to assess the lipophilicity of the compounds (Table S1). All the values of **D1-3** was higher than those of **P1-3**. The calculated LogP values (ALogP) and retention time on the octadecyl column of **D1-3** were also higher than those of **M1-3** although the $\log D_{\text{dec/w}}$ values of **D1-3** were not very different from those of **M1-3**. These results indicate that ester is more

lipophilic than amide and can be also more lipophilic than *N*-methanamide.

Table S1. Permeability values (P_e), ALogP, UPLC retention time, and $\log D_{\text{dec/w}}$ of **P1–3**, **D1–3**, and **M1–3**.

Amide					
Name	(Xaa ₁ , Xaa ₂)	P_e ($\times 10^{-6}$ cm/s)	ALogP	UPLC retention time (min)	$\log D_{\text{dec/w}}$
P1	(Phe, Phe)	0.07	1.34	7.67	-2.75 ± 0.01
P2	(Leu, Phe)	0.04	1.01	7.44	-3.24 ± 0.01
P3	(Leu, Leu)	0.01	0.69	6.84	-3.57 ± 0.03

Amide-to-ester substitution					
Name	(Xaa ₁ , Xaa ₂)	P_e ($\times 10^{-6}$ cm/s)	ALogP	UPLC retention time (min)	$\log D_{\text{dec/w}}$
D1	(Phe, Phe)	1.2	1.98	9.07	-1.17 ± 0.01
D2	(Leu, Phe)	0.4	1.66	8.63	-2.0 ± 0.1
D3	(Leu, Leu)	0.2	1.33	8.25	-2.142 ± 0.001

Backbone N-methylation					
Name	(Xaa ₁ , Xaa ₂)	P_e ($\times 10^{-6}$ cm/s)	ALogP	UPLC retention time (min)	$\log D_{\text{dec/w}}$
M1	(Phe, Phe)	0.3	1.54	8.74	-1.24 ± 0.01
M2	(Leu, Phe)	0.2	1.22	8.36	-1.62 ± 0.02
M3	(Leu, Leu)	0.1	0.89	7.63	-2.13 ± 0.02

Comment from the reviewer

- Section 2 -Cyclic Peptides

Figure 2c and 2d- Again no statistical analysis is provided. Are all 5 comparisons statistically significant?

>Our response to the comment

We have added statistical analysis in Figure 2c and 2d. The statistical analysis showed that there are significant differences.

[Revised manuscript, Figure 2c and 2d]

Figure 2. (c) PAMPA and (d) Caco-2 assay of synthesized cyclic peptides. PAMPA was conducted with 2 μM compounds in PBS containing 5% DMSO and 16 h incubation at 25 $^{\circ}\text{C}$. Cyclosporin A (CSA) was included as a control for PAMPA (0.4×10^{-6} cm/s). Each bar represents mean value and standard deviation from experiments carried out in quadruplicate. Caco-2 assay was conducted with 1 μM compounds in HBSS (pH 7.4) containing 10 mM HEPES and 1% DMSO and 3 h incubation at 37 $^{\circ}\text{C}$. Each bar represents mean value and standard deviation from experiments carried out in triplicate or quadruplicate. The statistical significance of **DP1–5** against **CP1** is shown above the bar of **DP1–5** and the statistical significance of **DP1–5** against **MP1–5** is shown above the bars of **DP1–5** and **MP1–5**. ** $p < 0.01$, * $p < 0.05$. n.s. denotes “not significant”.

Comment from the reviewer

- Figure 2f- No error bars at all are on this curve CP50 3.4 vs 7.4 μM are reported with no error of measurement or statistical significance. Are these numbers actually different?

>Our response to the comment

The error bars were shown on Figure 2f, but they were almost invisible because of the faint color. Therefore, we changed the color. Besides, we have provided the standard deviation and the information about the statistical significance on Figure 2e. The difference of CP_{50} values is indeed statistically significant.

Figure 2. (e) Chemical structure, linkages, and CP₅₀ values of chloroalkane-tagged cyclic peptides. CP₅₀ values, the concentration at which 50% cell penetration was observed, are shown at the bottom. (f) The results of CAPA for **CP1-L2Kct** (grey) and **DP1-L2Kct** (orange) analyzed by flow cytometry. Each data point represents the mean value of experiments carried out in triplicate and the error bars represent standard deviations of the triplicate.

Comment from the reviewer

- Figure 2g- The confocal microscopy appears to show a much larger difference than the flow cytometry results, which are virtually identical. Why is this the case?

>Our response to the comment

Please note that the confocal microscopic observation was performed after treatment of the cells with 5 μM peptides (the information of the concentration was added to the figure caption of Figure 2g.) According to the flow cytometric analysis shown in Figure 2f, the normalized fluorescence of the cells treated with **CP1-L2Kct** is about 65% while that of the cells treated with **DP1-L2Kct** is about 40%. When we analyzed the microscopic images in Figure 2g, the fluorescence intensity of the cells treated with **CP1-L2Kct** was about 61% of the vehicle control while the intensity of the cells treated with **DP1-L2Kct** was found to be about 37% of the images of the vehicle control (Reviewer only Figure). The two results (flow cytometric analysis and microscopic observation) are consistent with each other. The analysis of the flow cytometric result and microscopic images are shown as a Reviewer only Figure below as a reviewer-only material.

Reviewer only Figure. Comparison of the microscopic images (top) and the flow cytometric analysis (bottom).

Comment from the reviewer

- The explanation of structural heterogeneity for DP2 (from line 132 to line 142) by NMR spectroscopy is inconclusive and confusing. Multiple sets of NMR signals for Tyr-1 could be due to impurity. If multiple sets of NMR signals are the result of exchangeable structures of DP2 in solution, then variable temperature and/or solution NMR experiments should be carried out to experimentally support this claim.

- Another confusing statement from the authors is that there are three sets of NMR resonances for Tyr-1 H_n and H_{cb}, but only two sets of peaks for H_{ca} (lines 137 to 138). The H_n and H_{ca} of Tyr-1 should have the same number of sets of signals.

>Our response to the comment

We appreciate the reviewer's critical comment. After the reviewer's comment, we have resynthesized **DP2** and conducted the NMR measurements using freshly prepared samples. As a result, only a single set of peaks was observed. This indicates that the multiple sets of peaks of the Tyr-1 residue observed on the originally submitted manuscript were derived from modification of **DP2** or impurities.

Using the newly obtained NMR spectra, we have determined the stable conformation of **DP2** in CDCl₃. The **DP2** has been shown to have a different intramolecular hydrogen bonding network from that of **CP1**. One of the solvent-exposed amide hydrogen in **CP1** was found to form a new hydrogen bond in **DP2**. This may explain why **DP2** shows higher membrane permeability than **CP1**.

Therefore, we have removed the sentences describing the conformational heterogeneity of **DP2** from the manuscript. Besides, we have added sentences describing the stable conformation of **DP2** and the comparison of the conformation with the stable conformation of **CP1** as follows. These changes do not influence the main conclusions of the manuscript.

[Revised manuscript, p.9]

From the NMR spectra in CDCl₃, stable conformations of **DP2** in the lipophilic environment were determined. Interestingly, the amide NH of Tyr-1 residue in the most stable conformations of **DP2** forms an intramolecular hydrogen bond with the carbonyl oxygen of the Leu-5 residue (Figure S7a) while the same amide NH was reported to be solvent-exposed in **CP1**³² (Figure S7b). The amide-to-ester substitution on the amide that forms an intramolecular hydrogen bond in **CP1** caused the rearrangement of the intramolecular hydrogen bonding network, which is assumed to be the reason for the unexpectedly improved membrane permeability of **DP2**. The unexpected improvement of membrane permeability of **DP3** and **DP4** is also probably due to the conformational changes upon the amide-to-ester substitution as seen in **DP2**.

Figure S7. Comparison of the most stable conformations of DP2 and CP1.

The most stable conformations of (a) **DP2** and (b) **CP1** from NMR analysis are shown. Intramolecular hydrogen bonds are highlighted by dotted blue lines. Shown in (b) is the most stable conformation from our NMR analysis (Figure 3a and Figure S9–14), which is similar to the reported conformation in the previous report (Rezai, T., *et al.*, *J. Am. Chem. Soc.* **128**, 2510–2511 (2006)).

Comment from the reviewer

- There are inconsistent results for CP1 cell permeability as figure 2c and 2d indicated CP1 has very low membrane and cell permeability. However figure 2e, 2f and 2g showed CP1 with chloroalkane tag can significantly penetrate through the cell membrane, is this result due to the tag itself entering the cell or enhancing permeability?

>Our response to the comment

As the reviewer suggested, the difference of the improvement of membrane permeability of **DP1** between PAMPA/Caco-2 assay and CAPA may be caused by the effect of chloroalkane tag. The direct comparison of PAMPA/Caco-2 and CAPA is difficult because PAMPA and Caco-2 assay are methods to measure the rate of membrane permeation, whereas the CAPA assay is a method to measure %peptide inside the cells after a certain incubation time with a particular concentration of a peptide. The difference of the results from the different assays may be also derived from the difference of the carrier to be permeated in each assay. CAPA measures the efficiency of translocation into cytosol, whereas Caco-2 assay measures the permeability through cellular monolayer, and PAMPA measures the permeability through artificial membranes.

We have described the possible reasons for the difference among the assays in the manuscript as follows.

[Revised manuscript, p.11]

The difference in permeability between **CP1** and **DP1** was not as large as the difference observed in the PAMPA and Caco-2 assay, which could be because the chloroalkane tag affects the permeability of the peptides and/or there is a difference among the permeation process across the cell membrane, artificial membrane, and cell monolayer.

Comment from the reviewer

- Section 3-

There are no NOE summaries, amide coupling constants, variable temperature amide chemical shifts included for solving 3D structures in the SI. These data are absolutely essential for investigating and verifying any solution structure (e.g. CP1, DP1 and MP1), and for reviewing this paper.

>Our response to the comment

We have summarized NOEs and amide coupling constant in supporting information files.

Moreover, we conducted amide temperature coefficient measurements of **CP1**, **DP1**, and **MP1** in deuterated chloroform. Based on the obtained $\Delta\delta_{\text{NH}}/\Delta T$ values of **CP1**, we have modified the discussion about the conformational states of **CP1**, **DP1**, and **MP1** as follows.

Considering that the conformations derived from NMR measurements are average solution structures and the cyclic peptide structure dynamically changes, it is not meaningful and rather imprecise to depict specific intramolecular hydrogen bonding networks in the snapshot structure. Therefore, we deleted the highlight of hydrogen bonding networks in Figure 3.

[Revised manuscript, p.14]

As discussed in the previous section, the higher membrane permeability of **DP1** and

DP5 than that of **CP1** is presumably because an exposed amide NH of **CP1** in the membrane is removed upon the amide-to-ester substitution. To evaluate the validity of this assumption, we conducted a conformational analysis of **DP1** in CDCl₃ which mimics the environment in membrane. First, we reproduced the NMR structure of **CP1**. A similar conformation with that previously reported was obtained as the most stable conformation (Figures 3a and S9–14).³² In this conformation, amide hydrogen of Tyr-1 is exposed to solvent. In consistent with the previous report, amide hydrogen of Leu-5 residue is not involved in intramolecular hydrogen bonding in the most stable conformation, but the amide hydrogen faces inward in the molecule and probably partially masked from the solvent.

The conformational states of **DP1** in CDCl₃ were investigated using the same procedure as that of **CP1**. The superposition of **DP1** and **CP1** with their backbones showed that the most stable conformation of **DP1** is similar to that of **CP1** (Figure 3c). The root-mean-square deviation (RMSD) of their backbones was calculated to be 0.426, which confirmed that the amide-to-ester substitution of the exposed amide NH does not significantly change the solution conformations of the cyclic hexapeptide in a membrane-like lipophilic environment. Therefore, the substitution reduces the total number of solvent-exposed amide hydrogens, leading to improved membrane permeability.

We next investigated the solution NMR structure of **MP1**. The RMSD between the backbones of **MP1** and **CP1** was 0.782. The value is small but higher than the RMSD between **DP1** and **CP1**. These results indicate that **MP1** forms a conformational state that is similar but a little different from that of **CP1** in a lipophilic environment (Figures S21–27).

One of the possible reasons for the higher permeability of **DP1** than **MP1** can be the difference in the conformations between **DP1** and **MP1** in membrane. The NMR-based conformational analysis suggests that an amide-to-ester substitution does not largely change the conformation of **CP1** and removed a solvent-exposed amide NH of Tyr-1 residue while an amide *N*-methylation changed the conformation of **CP1** and did not reduce the total number of solvent-exposed amide NHs in a low dielectric environment. The conformational aspects were further assessed by amide temperature coefficient ($\Delta\delta_{\text{NH}}/\Delta T$) measurements (Table S3). While the trends in $\Delta\delta_{\text{NH}}/\Delta T$ values of Leu-2–Leu-5 residues of **CP1** and **DP1** are similar, that of **MP1** largely differ from

those values of **CP1** and **DP1**. Of note, the $\Delta\delta_{\text{NH}}/\Delta T$ value of D-Leu-3 residue in **MP1** is smaller than -4.6 ppb/K, suggesting that the amide NH is exposed to solvent.¹³

Another possible reason for the higher membrane permeability of **DP1** than **MP1** is that the lipophilicity of an ester bond is higher than that of an *N*-methylamide bond as suggested from the model dipeptide study. We calculated CLogP values to estimate the lipophilicity of these peptides. **MP1** has the highest CLogP (8.11), followed by **DP1** (7.52), and **CP1** (7.46). We also calculated ALogP values to estimate the lipophilicity of these peptides, which is a measure of compound lipophilicity calculated using a regression model based on the sum of the atomic lipophilicity of compounds. According to a previous report, ALogP value is a more accurate predictor of lipophilicity than CLogP for molecules with more than 45 atoms.³⁸ **DP1** has the highest ALogP (4.44), followed by **MP1** (4.00), and **CP1** (3.80). Since three-dimensional structures are not considered in these calculated lipophilicity values, the calculations suggest that the higher lipophilicity of an ester bond than that of an *N*-methylamide bond is one reason for the higher permeability of **DP1** than **MP1**.

Based on these results, the higher membrane permeability of **DP1** than **MP1** is assumed to be because **DP1** is more stable than **MP1** in a membrane-like low dielectric environment due to the smaller number of solvent-exposed amide NHs and the higher local lipophilicity of an ester bond. The experimentally determined 1,9-decadiene–water distribution coefficient $\log D_{\text{dec/w}}$ was consistent with the assumption: **DP1** exhibited the highest $\log D_{\text{dec/w}}$ (1.2), followed by **MP1** (0.052), and **CP1** (-0.74).

[Revised Figure 3]

Figure 3. Stereoviews of NMR solution structures of (a) **CP1** and (b) **DP1** in CDCl_3 . (c) The superposition of **DP1** with **CP1**. **DP1** is shown in brown and **CP1** is shown in blue.

Comment from the reviewer

- Figure S11. ROESY spectrum for CP1 does not show the crucial inter-residue ROEs

>Our response to the comment

In Figure S14 of the revised manuscript, observed inter- and intra-residue ROE signals are shown with their assignments, and listed in a supplementary Excel file. The observed correlations contain inter-residue ROEs, such as H_{NH} of Tyr-1 and $H_{C\alpha}$ of D-Pro-6.

Comment from the reviewer

- Figure S17. ROESY spectrum for DP1 shows overlapping alpha proton signals for residues l3, L4 and p6. There was no discussion of how the authors derived ROEs from those overlapping signals. This ROESY spectrum also does not show any inter-residue ROEs, suggesting that the structure derived from it is unreliable.

>Our response to the comment

The ${}^3H_{C\alpha}$ -L4 $H_{C\alpha}$ and L4 $H_{C\alpha}$ -p6 $H_{C\alpha}$ chemical shift differences of **DP1** in $CDCl_3$ are as small as 10 and 12 Hz, respectively, as the reviewer described. However, the chemical shift differences are sufficiently larger than the resolution of the FID and interferogram of the ROESY experiment (4 and 8 Hz, respectively), thus the signals can be separated in this study. The enlarged figure of the ROESY spectrum of **DP1** is shown below for the reviewer's convenience.

As shown in the ROESY spectrum and the NOE summary table, there are many inter-residue ROEs that helped the NMR structure calculation.

Reviewer Only Figure. An enlarged view of the ROESY spectrum. The cross peaks involving p6, L4 and l3 are clearly separated.

Comment from the reviewer

Only AlogP was used to calculate predicted ‘lipophilicity’ - why not any other calculations? Do other calculations lead to the same predictions for amide to ester replacements? AlogP often does not accurately predict experimental logP data.

>Our response to the comment

Thank you for the critical comment. We adopted ALogP value because ALogP value was suggested to be a more accurate predictor of lipophilicity than CLogP for molecules with more than 45 atoms in a previous report (Ghose, A. K., *et al.*, *J. Phys. Chem. A* **102**, 3762–3772 (1998)). However, according to the reviewer’s comment, we have also calculated CLogP values of **CP1**, and its derivatives with amide-to-ester substitution and *N*-methylation. **MP1** has the highest CLogP (8.11), followed by **DP1** (7.52), and **CP1** (7.46).

We have included the CLogP values in the manuscript and discussed lipophilicity using CLogP, AlogP and experimentally determined $\log D_{dec/w}$ values in the manuscript as follows.

[Revised manuscript, p16]

Another possible reason for the higher membrane permeability of **DP1** than **MP1** is that the lipophilicity of an ester bond is higher than that of an *N*-methylamide bond as suggested from the model dipeptide study. We calculated CLogP values to estimate the lipophilicity of these peptides. **MP1** has the highest CLogP (8.11), followed by **DP1** (7.52), and **CP1** (7.46). We also calculated ALogP values to estimate the lipophilicity of these peptides, which is a measure of compound lipophilicity calculated using a regression model based on the sum of the atomic lipophilicity of compounds. According to a previous report, ALogP value is a more accurate predictor of lipophilicity than CLogP for molecules with more than 45 atoms.³⁸ **DP1** has the highest ALogP (4.44), followed by **MP1** (4.00), and **CP1** (3.80). Since three-dimensional structures are not considered in these calculated lipophilicity values, the calculations suggest that the higher lipophilicity of an ester bond than that of an *N*-methylamide bond is one reason for the higher permeability of **DP1** than **MP1**.

Based on these results, the higher membrane permeability of **DP1** than **MP1** is assumed to be because **DP1** is more stable than **MP1** in a membrane-like low dielectric environment due to the smaller number of solvent-exposed amide NHs and the higher local lipophilicity of an ester bond. The experimentally determined 1,9-decadiene–water distribution coefficient $\log D_{\text{dec/w}}$ was consistent with the assumption: **DP1** exhibited the highest $\log D_{\text{dec/w}}$ (1.2), followed by **MP1** (0.052), and **CP1** (−0.74).

Comment from the reviewer

Section 4-

The *in silico* simulation is the strongest part of this work. This is why this work may be better suited for a computational chemistry journal.

>Our response to the comment

Indeed, the *in silico* simulation is one of the important parts of this paper. This is because the simulations have for the first time provided insights on how backbone amide modifications affect the membrane permeation process of cyclic peptides. During the revision period, we have conducted NMR measurements to obtain

experimental evidence for the MD simulations. The NMR experiments are consistent with the MD simulations as described in our response to the reviewer's next comment.

The conformational changes upon membrane permeation suggested by simulations are now supported by experiments. Therefore, this work provides new insights into how cyclic peptides permeate lipid bilayer, how the permeation process is affected by backbone modifications, and clues for designing membrane-permeable peptides, a growing topic in biological and medical sciences. Therefore, we believe this work is of great interest to a broad audience which is not limited to computational scientists.

Comment from the reviewer

It is crucial that modeling results in this section are supported by at least some experimental data (e.g. CD, NMR amide proton temperature dependence, amide coupling constants to name just a few). Modeling predictions are notoriously prone to erroneous interpretations when not supported by strong experimental evidence of structure to prove the model.

>Our response to the comment

We appreciate the reviewer's constructive criticism.

As we have discussed in the manuscript, the conformations of **CP1**, **DP1**, and **MP1** in membrane (in the environment with a low dielectric constant) derived from the steered MD simulations are consistent with the conformations determined by NMR analysis. On the other hand, the conformations of the peptides in water (in the environment with a low dielectric constant) derived from the MD simulations had not been experimentally examined in the originally submitted manuscript.

To assess the validity of the MD simulations in water, we have conducted the NMR experiments of **CP1**, **DP1**, and **MP1** in water containing 50% DMSO as a solvent with a high dielectric constant.

The experimental data have suggested that all the three peptides form open conformations in a solvent with a high dielectric constant, which is consistent with the enhanced sampling MD simulations in an aqueous solution. This indicates the validity of the conformational changes of the cyclic peptides during the membrane

permeation process suggested by the MD simulations.

The following sentences and figure have been added in the revised manuscript.

[Revised manuscript, p20]

The conformational changes of the cyclic peptides during the process of membrane permeation suggested from the MD simulations were experimentally examined. In the membrane, the conformer A, which was the representative of the dominant conformational states, was similar to the conformations of **CP1**, **DP1**, and **MP1** determined by the NMR measurements in CDCl₃ (RMSD of their backbone was calculated to be 0.260, 0.433, and 0.908 Å for **CP1**, **DP1**, and **MP1**, respectively). In order to assess the validity of the conformations in an aqueous solution, NMR spectra of **CP1**, **DP1**, and **MP1** in a solvent with a high dielectric constant were measured (Figure S28–46). 1:1 mixture of DMSO and water was used as the solvent. DMSO was added to fully solubilize the peptides. The following two lines of evidence obtained from the NMR analysis suggested the validity of the conformations observed in an aqueous solution in the enhanced sampling MD simulations. First, for all three peptides, the number of inter-residue NOE signals was smaller in the high-dielectric solvent than in CDCl₃, suggesting that the peptides form more “open” conformations with a smaller number of intramolecular hydrogen bonds. Second, all the pairs of protons that gave NOE signals have average distances of less than 5 Å in the enhanced sampling MD simulations. When the NMR-derived conformations and the representative conformations from the enhanced sampling MD simulations in the aqueous environment (conformer C for **CP1** and **DP1**, and conformer E for **MP1**) are compared (Figure S28), the RMSD value was 0.578 Å, 1.188 Å, 0.861 Å for **CP1**, **DP1**, and **MP1**, respectively, suggesting the consistency of the simulations and experiments. The RMSD is a little high for **DP1** and the conformation from the enhanced sampling MD simulations was more open than that from the NMR analysis. This can be explained by the fact that the simulations were conducted in water while NMR measurements were conducted in 1:1 mixture of DMSO and water which has a smaller dielectric constant than that of pure water. Altogether, the NMR-based conformational analysis validates the conformational changes during the membrane permeation process suggested by the enhance sampling MD simulations.

Figure S28. Comparison of the representative conformations in high dielectric solvents from NMR and enhanced sampling MD simulations.

The overlay of the representative conformations of (a) **CP1**, (b) **DP1**, (c) **MP1** from NMR in 1:1 mixture of DMSO and water (green) and simulations in water (cyan). Two structures are overlaid using backbone atoms (N, C α , C', and O).

Comment from the reviewer

Section 5-

I find the title of this section misleading as only cyclic hexapeptides with the sequence [PLLLLY] were used. As such the authors' claims of being able to improve permeability in a range of peptides is misleading and actually pretty limited.

>Our response to the comment

In this section, we used [PLLLLY] with different stereochemistries to understand the scope and limitation of amide-to-ester substitution on the peptides. However, the reviewer's criticism is reasonable because the sequence or ring size of the peptide is unchanged in the originally submitted manuscript.

To further evaluate the scope and limitation of the amide-to-ester substitution strategy for improving membrane permeability of cyclic peptides, we have investigated the effect of an amide-to-ester substitution on membrane permeability of cyclic hexapeptides with different sequences and cyclic peptides with larger ring size. The substitution on the cyclic hexapeptides and cyclic 8- and 9-mer peptides also improved membrane permeability on PAMPA/Caco-2 assay. The result indicates that introduction of an amide-to-ester substitution has a potential to improve membrane

permeability of cyclic peptides, not only hexapeptides [PLLLLY], but also cyclic peptides with different sequences and larger ring sizes.

With the newly added experimental results, we believe the title of the section “The Effect of Amide-to-ester Substitution on Membrane Permeability of Other Cyclic Peptides” is deemed suitable.

The newly obtained permeability data of other cyclic peptides are described in the earlier response to another comment shown on p.35–37 of this document.

Comment from the reviewer

No *n*-methyl comparisons were used in this section.

>Our response to the comment

We have evaluated the membrane permeability of *N*-methyl analogs of **CP2–6** and included the results in Figure S48 as shown below. In all the cases, the membrane permeability of cyclic peptides with an amide-to-ester substitution has shown comparable or better membrane permeability than the corresponding peptides with an amide *N*-methylation.

Figure S48. PAMPA of cyclic hexapeptides CP2–CP6 and their derivatives

(a) General chemical structure of CP2–6 and (b) backbone stereochemistries of CP2–6.⁶ The backbone stereochemistries on Leu-3, Leu-4, and Leu-5 are different among the compounds. (c) The position of the introduction of an amide-to-ester substitution or an amide *N*-methylation. (d) The permeability of compounds measured by PAMPA. 3 μ M compounds dissolved in 5% DMSO/PBS (pH 7.4) were incubated in donor wells docked with acceptor wells containing 5% DMSO/PBS (pH 7.4) for 16 h at 25 °C. After the incubation, the concentration of compounds in each well was measured by LC-MS to calculate their permeability. ** $p < 0.01$, * $p < 0.05$, n.s. denotes “not significant”.

Comment from the reviewer

This paper would be greatly enhanced if ester replacements were made to a broader range of peptides and the effects on permeability and structure explored.

>Our response to the comment

We appreciate the encouraging comment. We have provided new data about cyclic hexapeptides with different sequences and cyclic 8- and 9-mer peptides as described in our response to the earlier questions (p.35–37 of this document). We hope the new result demonstrates the broad scope of the amide-to-ester substitution on improving membrane permeability of various cyclic peptides.

Comment from the reviewer

Again no statistical analysis.

>Our response to the comment

We have added the information about the statistical significance in Figure S48d in the revised supporting information. Figure S48d is shown in the previous page of this document.

Comment from the reviewer

Section 6-

Stability of peptides in different pH buffers should also be examined, since depsipeptides are well known to be prone to hydrolysis. Indeed, that is a major reason why depsipeptides have not translated well in medicinal chemistry - in most cases they tend to be quite unstable in vivo and prone to cleavage.

>Our response to the comment

We appreciate the constructive recommendation. As the reviewer described, the environment in vivo is not only neutral, but also acidic. Especially, the pH in gastric environment is extremely low. According to the reviewer's comment as well as one of the comments from reviewer 1 in which evaluation of the peptide stability in simulated gastric fluid (SGF) as well as in simulated intestinal fluid (SIF) is recommended, we have evaluated the stability of **CP1**, **DP1**, and **MP1** in SGF and SIF. pH value in SGF and SIF is 1.2 and 6.8, respectively. As a result, the depsipeptide was stable in SGF and SIF for 4 h. Together with the result of stability assay in serum

and plasma which is conducted in neutral buffers, these results suggest decapeptide is stable in solutions with pH values typical *in vivo*.

[Revised Figure 6c and d]

Figure 6. (c) Stability in simulated gut fluid. 98% of a control peptide (somatostatin) was degraded at 4 h under the same conditions. (d) Stability in simulated intestinal fluid. 94% of a control peptide (oxytocin) was degraded at 4 h under the same conditions. Degradation profiles of the control peptides are shown in Figure S49. In (b)–(d), each point represents mean value and standard deviation from experiments carried out in triplicate.

Figure S49. Degradation profiles of control peptides in SGF and SIF

(a) Stability of somatostatin in simulated gut fluid (SGF). (b) Stability of oxytocin in simulated intestinal fluid (SIF).

Comment from the reviewer

Discussion

Figure 6 - this figure is an oversimplification, the effect will not be just because of one water interaction.

>Our response to the comment

We thank the reviewer's comment. In Figure 6 of the initially submitted manuscript, we intended to illustrate that the removal of a solvent-exposed amide proton by an amide-to-ester substitution reduces the energetic cost for the cyclic peptide to go inside the membrane. However, as the reviewer described, drawing the one-to-one hydrogen bond between a water molecule and the amide hydrogen is an oversimplification. Therefore, we have removed the figure from the manuscript.

Comment from the reviewer

Methods

PAMPA assay: Different assay systems were used for CP1 and derivatives than for the other peptides. Why?

The Genetest Pre-coated PAMPA plate can give very different results than the homemade system. All compounds need to be measured in the same system to make valid comparisons.

>Our response to the comment

Thank you for carefully reviewing the experimental methods. After receiving the reviewer's comment, we have considered that all the assays should be conducted with the same PAMPA plate. Therefore, we have re-conducted PAMPA of **CP1**, **DP1–P5**, and **MP1–P5** again using the PAMPA plate used for other compounds in the manuscript. As a result, the same trend was observed for the permeability of the compounds. The data in Figure 2c was replaced with the newly obtained data.

Figure 2. (c) PAMPA and (d) Caco-2 assay of synthesized cyclic peptides. PAMPA was conducted with 2 μ M compounds in PBS containing 5% DMSO and 16 h incubation at 25 °C. Cyclosporin A (CSA) was included as a control for PAMPA (0.4×10^{-6} cm/s). Each bar represents mean value and standard deviation from experiments carried out in quadruplicate.

Comment from the reviewer

NMR measurements: Spectra are recorded in CDCl_3 but DMSO appears in some of the spectra. Does the presence of this more polar solvent affect the structures presented?

>Our response to the comment

Thank you for carefully reviewing our spectroscopic data. Because the compound was originally stored in DMSO after synthesis, a trace amount of DMSO remained after lyophilization and observed in the NMR spectra. It is possible that the residual DMSO affect to the conformation of the cyclic peptide, thus we have measured NMR spectra of **CP1**, **MP1** and **DP1** again. To avoid the contamination of DMSO, we have directly dissolved the compounds with CDCl_3 . The stable conformations derived from the newly obtained NMR spectra are consistent with the conformations obtained from the previous analysis. We have replaced the spectra on the Supporting Information with the newly obtained spectra. We have also replaced the stable conformations shown in Figure 3 with the newly obtained ones. Because there are no large differences between the conformations on the originally submitted manuscript and on the revised manuscript, the changes do not affect the main conclusions of the manuscript.

Comment from the reviewer

- Authors: Why so many corresponding authors?

>Our response to the comment

Each corresponding author is responsible as follows.

Shinsuke Sando and Jumpei Morimoto: The research has been initiated at the University of Tokyo, and most experiments have been conducted in their lab.

Scott Lokey: His lab has been studying the effect of amide-to-ester substitution on cyclic peptides independently from UTokyo group. During a meeting of UTokyo group and Lokey group, they found that both groups are studying amide-to-ester substitution and decided to collaborate on this work.

Yutaka Akiyama: His group has been developing novel MD simulation methodologies to study conformational changes of cyclic peptides during membrane permeation process. As the reviewer described in the review comment, the MD simulation part is one of the highlights of this manuscript. The MD simulation methodologies, independently developed by his group, were the key to realize the analysis of the conformational changes. He is especially responsible for the computational part.

REVIEWERS' COMMENTS

Reviewer #1 (Remarks to the Author):

The authors have performed additional experiments to justify the referees' comments, significantly improving the manuscript. I am satisfied with their responses. However, a few minor comments should be addressed before acceptance of the manuscript for publication.

a) On page 27, the authors suggest using hydrophilic peptides to assess the applicability of their method. However, it is difficult to predict if CP1-Y1F-L2S is significantly more polar than CP1. Are the LogD7.4 values of CP1-Y1F-L2S/L2K very different than CP1?

b) In Fig 5b, the y-axis scales on the left and right are different. This somehow should be highlighted or mentioned in the Fig legend.

c) On page 34, "substituting an amide bond exposed in a lipophilic environment" is confusing. An amide bond has an "O" and "H" component that is exposed and shielded (or vice versa).

d) It is surprising to note the complete protection of CP1 in SGF and SIF. That also suggests no benefit of the chemical modification on the cyclic peptide scaffold. Were the simulated fluids prepared per USP guidelines, and were their activities measured?

Reviewer #2 (Remarks to the Author):

The authors have addressed my comments.

Reviewer 1

Original comments from Reviewer 1

The authors have performed additional experiments to justify the referees' comments, significantly improving the manuscript. I am satisfied with their responses. However, a few minor comments should be addressed before acceptance of the manuscript for publication.

a) On page 27, the authors suggest using hydrophilic peptides to assess the applicability of their method. However, it is difficult to predict if CP1-Y1F-L2S is significantly more polar than CP1. Are the LogD_{7.4} values of CP1-Y1F-L2S/L2K very different than CP1?

b) In Fig 5b, the y-axis scales on the left and right are different. This somehow should be highlighted or mentioned in the Fig legend.

c) On page 34, "substituting an amide bond exposed in a lipophilic environment" is confusing. An amide bond has an "O" and "H" component that is exposed and shielded (or vice versa).

d) It is surprising to note the complete protection of CP1 in SGF and SIF. That also suggests no benefit of the chemical modification on the cyclic peptide scaffold. Were the simulated fluids prepared per USP guidelines, and were their activities measured?

A point-by-point response to the comments of Reviewer 1

We thank the reviewer for the careful reading of our revised manuscript. We provide answers to all the reviewer's comments as follows.

Comment from the reviewer

a) On page 27, the authors suggest using hydrophilic peptides to assess the applicability of their method. However, it is difficult to predict if CP1-Y1F-L2S is significantly more polar than CP1. Are the LogD7.4 values of CP1-Y1F-L2S/L2K very different than CP1?

>Our response to the comment

As the reviewer pointed out, it is not clear whether **CP1-Y1F-L2S** and **CP1-Y1F-L2K** are more hydrophilic than **CP1**. Therefore, we have calculated ALogP (simulated values of LogD7.4) of **CP1-Y1F-L2S/L2K** and found that the values are smaller than that of **CP1**, indicating that the two peptides are more hydrophilic than **CP1** as expected. The calculated values and the related discussion have been added to the revised manuscript as follows.

[Revised manuscript, p. 21]

CP1-Y1F-L2S and **CP1-Y1F-L2K** have smaller ALogP values (1.94 and 1.75, respectively) than that of **CP1** (3.80), suggesting the lower lipophilicity of the two peptides compared with **CP1**.

Comment from the reviewer

b) In Fig 5b, the y-axis scales on the left and right are different. This somehow should be highlighted or mentioned in the Fig legend.

>Our response to the comment

The graphs can be misleading as the reviewer pointed out. Therefore, we have added the sentence "*Note that the scale of the y-axis is different between the left and right graphs.*" in the Figure 5b legend (p. 51 of the revised manuscript).

Comment from the reviewer

c) On page 34, "substituting an amide bond exposed in a lipophilic environment" is confusing. An amide bond has an "O" and "H" component that is exposed and shielded (or vice versa).

>Our response to the comment

On page 26 of the revised manuscript, we have modified the expression to “*substituting an amide bond whose hydrogen is exposed in a lipophilic environment*” to clarify which of the amide bond is exposed.

Comment from the reviewer

d) It is surprising to note the complete protection of CP1 in SGF and SIF. That also suggests no benefit of the chemical modification on the cyclic peptide scaffold. Were the simulated fluids prepared per USP guidelines, and were their activities measured?

>Our response to the comment

We thank the reviewer for critically checking the newly added SGF/SIF assay results.

We prepared the SGF and SIF solutions by referring to the protocol described in the *U. S. Pharmacopeia & National formulary* to follow the USP guidelines and the protocol described in a recent report (T. Kremsmayr, *et al.*, *J. Med. Chem.* **2022**, *65*, 6191–6206) with some modifications. This information has been included in the Methods section of the revised manuscript as follows.

[Revised manuscript, p. 35]

The stability assay of peptides in SGF was conducted according to United States Pharmacopeia & National Formulary (USP 25-NF 20)⁶⁷ and the previous report⁶⁸ with some modifications.

[Revised manuscript, p. 36]

The stability assay of peptides in SIF was conducted according to United States Pharmacopeia & National Formulary (USP 25-NF 20)⁶⁷ and the previous report⁶⁸ with some modifications.

The activities of the simulated fluids were checked by including somatostatin and oxytocin as a control for the SGF assay and SIF assay, respectively. The information about the control is included in Figure 6c and d legends and Figure S49. These are copied below for the reviewer's convenience.

[Revised manuscript, Figure 6c and d legends]

(c) Stability in a simulated gut fluid. 98% of a control peptide (somatostatin) was degraded at 4 h under the same conditions. (d) Stability in a simulated intestinal fluid. 94% of a control peptide (oxytocin) was degraded at 4 h under the same conditions. The degradation profiles of the control peptides are shown in Supplementary Figure 50.

[Revised SI, Supplementary Figure 50]

Supplementary Figure 50. Degradation profiles of control peptides in SGF and SIF

(a) Stability of somatostatin in simulated gut fluid (SGF). (b) Stability of oxytocin in simulated intestinal fluid (SIF).

Reviewer 2

Original comments from Reviewer 2

The authors have addressed my comments.

Our response to the comment of Reviewer 2

We thank the reviewer for reading our revised manuscript. We are glad to see that the reviewer was satisfied with our responses.